# An in silico FSHD muscle fiber for modeling DUX4 dynamics and predicting the impact of therapy

Matthew V Cowley[1†], Johanna Pruller[2†], Massimo Ganassi[2], Peter S Zammit[2], Christopher RS Banerji[2,3]*

[1]Centre for Sustainable and Circular Technologies, Department of Chemistry, University of Bath, Bath, United Kingdom; [2]King's College London, Randall Centre for Cell and Molecular Biophysics, New Hunt's House, Guy's Campus, London, United Kingdom; [3]The Alan Turing Institute, British Library, London, United Kingdom

**Abstract** Facioscapulohumeral muscular dystrophy (FSHD) is an incurable myopathy linked to the over-expression of the myotoxic transcription factor DUX4. Targeting DUX4 is the leading therapeutic approach, however, it is only detectable in 0.1–3.8% of FSHD myonuclei. How rare DUX4 drives FSHD and the optimal anti-DUX4 strategy are unclear. We combine stochastic gene expression with compartment models of cell states, building a simulation of DUX4 expression and consequences in FSHD muscle fibers. Investigating iDUX4 myoblasts, scRNAseq, and snRNAseq of FSHD muscle we estimate parameters including DUX4 mRNA degradation, transcription and translation rates, and DUX4 target gene activation rates. Our model accurately recreates the distribution of DUX4 and targets gene-positive cells seen in scRNAseq of FSHD myocytes. Importantly, we show DUX4 drives significant cell death despite expression in only 0.8% of live cells. Comparing scRNAseq of unfused FSHD myocytes to snRNAseq of fused FSHD myonuclei, we find evidence of DUX4 protein syncytial diffusion and estimate its rate via genetic algorithms. We package our model into freely available tools, to rapidly investigate the consequences of anti-DUX4 therapy.

**\*For correspondence:**
cbanerji@turing.ac.uk

[†]These authors contributed equally to this work

**Competing interest:** The authors declare that no competing interests exist.

## Editor's evaluation

To provide a logical answer to the over-expressed DUX4 in FSHD, the authors took a sophisticated mathematical modeling approach and applied it to empirical data. The approach successfully predicts behaviors of proteins and cells, thereby suggests a model for pathogenicity. The result poses a potential to be expanded to understand molecular dynamics of other mutation-mediated rare diseases.

## Introduction

FSHD is a prevalent (~12/100,000 *Deenen et al., 2014*), incurable, inherited skeletal myopathy. The condition is characterized by progressive fatty replacement and fibrosis of specific muscle groups driving weakness and wasting (*Banerji and Zammit, 2021*), which is accelerated by inflammation (*Dahlqvist et al., 2020*). FSHD is highly heterogeneous, with both the rate of progression and order of muscle involvement varying dramatically from person to person, even between monozygotic twins (*Tawil et al., 1993*). Around 75% of patients exhibit a descending phenotype with weakness beginning in the facial muscles, before progressing to the shoulder girdle and latterly lower limb, while the remaining 25% exhibit a range of 'atypical' phenotypes, including facial sparing and lower limb

predominant (*Banerji et al., 2020a*). Despite this clinical range, FSHD is associated with significant morbidity and socioeconomic costs (*Schepelmann et al., 2010*).

Genetically FSHD comprises two distinct subtypes: FSHD1 (OMIM: 158900, 95% of cases) and FSHD2 (OMIM: 158901, 5% of cases). Both subtypes bear a unifying epigenetic feature: derepression of the D4Z4 macrosatellite at chromosome 4q35. In FSHD1 this is due to truncation of the D4Z4 macrosatellite from the typical >100 to 10–1 units (*Lemmers et al., 2010*). In FSHD2, derepression is due to mutation in a chromatin modifier, typically *SMCHD1* (*Lemmers et al., 2012*) but rarely *DNMT3B* (*van den Boogaard et al., 2016*) or *LRIF1* (*Hamanaka et al., 2020*). In addition to D4Z4 epigenetic derepression, FSHD patients also carry certain permissive 4qA haplotypes distal to the last D4Z4 repeat encoding a polyadenylation signal (*Lemmers et al., 2010*).

Each 3.3 kb D4Z4 repeat encodes the transcription factor DUX4, which plays a role in zygotic genome activation, after which it is silenced in somatic tissues (*De Iaco et al., 2017*). In FSHD, however, epigenetic derepression of the D4Z4 region allows inappropriate transcription of DUX4 from the most distal D4Z4 unit, with transcripts stabilized by splicing to the polyadenylation signal in 4qA haplotypes, allowing translation. Mis-expression of DUX4 protein is thus believed to underlie FSHD pathogenesis and DUX4 inhibition is currently the dominant approach to FSHD therapy (*Tawil, 2020*; *Le Gall et al., 2020*).

However, DUX4 is extremely difficult to detect in FSHD patient muscle, with the vast majority of transcript and protein level studies failing to detect DUX4 in FSHD muscle biospies (*Banerji and Zammit, 2021*). When DUX4 is detected in FSHD patient muscle, it is at very low levels, requiring highly sensitive techniques such as nested RT-qPCR for transcripts (*Jones et al., 2012*) and proximity ligation assays for protein (*Beermann et al., 2022*). Investigation of FSHD patient-derived myoblasts has confirmed this very low level of DUX4 expression (*Banerji and Zammit, 2021*). Single-cell and single nuclear transcriptomic studies find only 0.5–3.8% of in vitro differentiated FSHD myonuclei express DUX4 transcript (*Jiang et al., 2020*; *van den Heuvel et al., 2019*). Immunolabelling studies only detect DUX4 protein in between 0.1–5% of FSHD myonuclei (*Snider et al., 2010*; *Rickard et al., 2015*).

As DUX4 is a transcription factor, it has been proposed that DUX4 target genes may represent a key driver of FSHD pathology. However, multiple meta-analyses have found DUX4 target gene expression to be a poor biomarker of FSHD muscle, except in the context of significant inflammation (*Banerji et al., 2017*; *Banerji and Zammit, 2019*), where it may be confounded by immune cell gene expression (*Banerji et al., 2020b*). Importantly, a recent phase 2b clinical trial of the DUX4 inhibitor losmapimod failed to reach its primary endpoint of reduced DUX4 target gene expression in patient muscle, despite improvement in functional outcomes (*Jagannathan et al., 2022*). Given the challenge of detecting DUX4 in muscle biopsies (*Banerji and Zammit, 2021*) it is unsurprising that no data was published relating to DUX4 expression changes during the losmapimod clinical trial. An understanding as to why losmapimod did not alter the expression of DUX4 targets in patient muscle biopsies is also lacking, but hypotheses include heterogeneity in muscle sampling, low baseline levels of DUX4 targets and thus limited dynamic range, slow reversibility of DUX4-induced epigenetic changes on target gene promoters and losmapimod having limited impact on DUX4 transcriptional activity in vivo.

How can such a rare expression of DUX4 drive a pathology as significant as FSHD? DUX4 expression in FSHD patient myoblasts follows a burst-like pattern, and cells expressing DUX4 have significantly shortened lifespans, suggesting a gradual attrition of cells over time (*Rickard et al., 2015*). A mouse model in which DUX4 expression is induced in a rare burst-like manner bears striking histological and transcriptomic similarities to FSHD patient muscle (*Bosnakovski et al., 2017*; *Bosnakovski et al., 2020*). DUX4 expression in FSHD patient-derived, multi-nucleated myotubes also displays a gradient across myonuclei, suggesting that DUX4 may 'diffuse' either actively or passively from an origin nucleus to neighboring nuclei, bypassing their need to wait for the rare DUX4 burst (*Rickard et al., 2015*; *Tassin et al., 2013*).

A deeper understanding of DUX4 dynamics and how it drives FSHD pathology is essential to move toward anti-DUX4 therapy. Not only would this explain DUX4 target gene expression as a suboptimal monitoring tool, but would enable optimal therapeutic design, through in silico perturbation of the parameters underlying DUX4 expression and toxicity.

Despite considerable discussion of 'rare-bursts' and 'diffusion' no attempt has been made to understand DUX4 expression through stochastic processes or differential equations; the natural setting to place these dynamics. Here, we combine ordinary differentiation equation models with stochastic gene expression models to construct a tuneable in silico simulation of DUX4 regulation in FSHD cell culture, both in unfused myocytes and syncytial multinucleated myotubes.

Through analysis of iDUX4 myoblasts, scRNAseq and snRNAseq of FSHD differentiated myoblasts we derive experimental estimates for the parameters of our model, including DUX4 mRNA degradation, transcription and translation rates, and DUX4 target gene activation rates. Simulation of our model provides a striking fit to DUX4 and DUX4 target gene expressing cell proportions seen in scRNAseq of FSHD myocytes. Importantly, our model predicts that DUX4 drives significant cell death, despite expression being limited to <1% of live cells. By comparing scRNAseq of unfused FSHD myonuclei to snRNAseq of multinucleated FSHD myotubes, we find evidence of DUX4 protein syncytial diffusion. We extend our model to examine DUX4 spreading between adjacent myonuclei and project our simulation onto the surface of a muscle fiber, in a spatially relevant model. Employing genetic algorithms to fit our spatial model to snRNAseq of FSHD syncytial myotubes, we provide an estimate for the syncytial diffusion rate of DUX4 protein.

We package our model into three freely available user interfaces, presenting an in silico toolkit to assess the impact of specific anti-DUX4 therapies on FSHD cell culture in a rapid, cost-effective, and unbiased manner.

## Results

### Compartment and promoter-switching models of DUX4 expression

Here, we consider two models of DUX4 expression in FSHD myocytes, a deterministic model of FSHD cell states we call the compartment model, and a stochastic model of gene expression we call the promoter switching model.

We first describe the compartment model. FSHD single myocytes can express DUX4 and, therefore, DUX4 target genes (*Rickard et al., 2015*; *Kowaljow et al., 2007*; *Heher et al., 2022*). We have previously demonstrated, via a library of DUX4 expression constructs, that DUX4 target gene activation is also necessary for DUX4-induced cell death (*Knopp et al., 2016*). We thus propose that an FSHD single myocyte at a given time $t$, occupies one of the following five states/compartments, defined by its transcriptomic distribution:

1. $S(t)$ – a *susceptible* state where the cell expresses no DUX4 mRNA and no DUX4 target mRNA (DUX4 -ve/Target gene -ve: DUX4 mRNA naive cell)
2. $E(t)$ – an *exposed* state where the cell expresses DUX4 mRNA but no DUX4 target mRNA (DUX4 +ve/Target gene -ve: DUX4 transcribed but not translated)
3. $I(t)$ – an *infected* state where the cell expresses both DUX4 mRNA and DUX4 target mRNA (DUX4 +ve/Target gene +ve: DUX4 transcribed and translated)
4. $R(t)$ – a *resigned* state where the cell expresses no DUX4 mRNA but does express DUX4 target mRNA (DUX4 -ve/Target gene +ve: i.e., a historically DUX4 mRNA expressing cell)
5. $D(t)$ – a *dead* state

We propose a model in which the single FSHD myocyte can transition through these five states according to five parameters:

1. $V_D$ – the average transcription rate of DUX4 in a single cell
2. $d_0$ – the average degradation rate of DUX4 mRNA
3. $T_D$ – the average translation rate of DUX4 from mRNA to active protein
4. $V_T$ – the average transcription rate of DUX4 target genes in the presence of DUX4
5. $D_r$ – the average death rate of DUX4 target positive cells

We allow cells to transition from DUX4 mRNA negative states $S(t)$ and $R(t)$ to DUX4 mRNA positive states $E(t)$ and $I(t)$, respectively at the rate of transcription of DUX4, $V_D$, with transition in the reverse direction occurring at the degradation rate of DUX4 mRNA, $d_0$. Transition from state $E(t)$ to state $I(t)$ requires DUX4 mRNA to be translated to active protein and DUX4 target mRNA to be expressed and thus occurs at the rate $T_D V_T$. Lastly, DUX4 target gene +ve states $I(t)$ and $R(t)$ transition to the dead

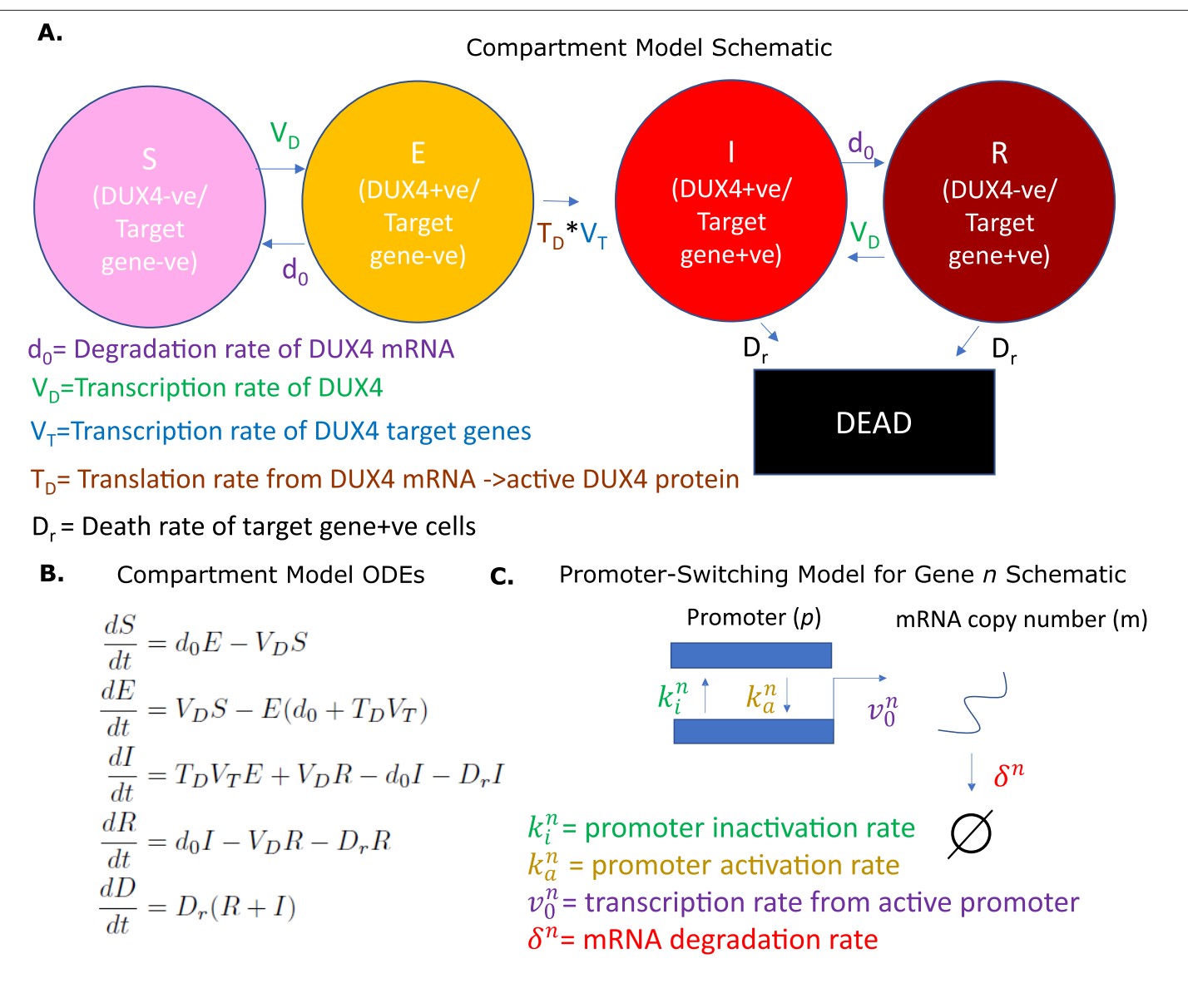

**Figure 1.** Overview of models. (**A**) Schematic of the compartment model describing the transition between five Facioscapulohumeral muscular dystrophy (FSHD) states according to five underlying parameters. (**B**) Ordinary differential equations describing the compartment model. (**C**) Schematic of the promoter-switching model of gene expression.

cell state $D(t)$ at rate $D_r$. Our compartment model can be represented schematically (*Figure 1A*) or equivalently as a system of ordinary differential equations (ODEs) (*Figure 1B*).

There are important assumptions in our compartment model:

1. Cells are assumed not to proliferate over the evolution of the model. We restrict applications to differentiating cells that have exited the cell cycle.
2. The death rate of DUX4 target gene -ve cells is negligible in comparison to the death rate of target gene +ve cells. This assumption is justifiable given published data on the death rate of DUX4 target gene-positive cells (*Rickard et al., 2015*).
3. The transition from DUX4 target gene -ve cell to DUX4 target gene +ve cell states is irreversible, i.e., we assume that the volume of target transcripts induced by DUX4 is sufficiently large, such that the rate of their degradation to zero is negligible in comparison to the death rate of target positive cells. This assumption is justified given that DUX4 is a potent transcriptional activator and pioneer factor (*Knopp et al., 2016*; *Choi et al., 2016*).

In what follows, we derive experimental estimates for the 5 parameters underlying the compartment model.

A further preliminary is the promoter-switching model. This model is precisely the two-stage telegraph process, which has a long history of study in stochastic gene expression (*Shahrezaei and Swain, 2008*; *Vu et al., 2016*; *Kim and Marioni, 2013*). Under this model, we assume that the promoter of a gene $n$ can occupy one of two states: active and inactive, and that transition from active (*a*) to inactive (*i*) state occurs at a rate $k_i^n$, with the reverse transition occurring at a rate $k_a^n$. We further assume that an active promoter can transcribe a single mRNA at a rate $v_0^n$, which degrades at a rate $\delta^n$. The model is represented schematically in *Figure 1C*. This model of gene expression has been shown to follow a Poisson-Beta distribution (*Vu et al., 2016*; *Kim and Marioni, 2013*), where the promoter state is determined by a Beta-distributed variable $p \sim Beta\left(k_a^n/\delta^n, k_i^n/\delta^n\right)$, and the mRNA copy number distribution conditional on the promoter state follows a Poisson distribution $mlp \sim Poisson\left(pv_0^n/\delta^n\right)$. Under this interpretation maximum likelihood estimates (MLEs) for the normalized underlying parameters $k_a^n/\delta^n$, $k_i^n/\delta^n$ and $v_0^n/\delta^n$ can be approximated from single-cell transcriptomic data (*Vu et al., 2016*; *Kim and Marioni, 2013*).

We assume that the proportion of time the promoter is in the active state, multiplied by the transcription rate of the active promoter in our promoter-switching model $\frac{k_a^n v_0^n}{k_a^n + k_i^n}$, is a reasonable proxy for the average rate of transcription of a gene in our compartmental model. We employ this assumption to estimate the average transcription rates $V_D$ and $V_T$ for the compartment model.

## Estimating the kinetics of DUX4 mRNA

Our compartment model contains two parameters governing the kinetics of DUX4 mRNA: the degradation rate $d_0$ and the average transcription rate $V_D$.

To estimate the degradation rate $d_0$ we employed human immortalized LHCN-M2 myoblasts expressing DUX4 under the control of a doxycycline-inducible promoter (iDUX4 myoblasts) (*Choi et al., 2016*). DUX4 expression was induced to a level and duration we have found sufficient to drive robust DUX4 mRNA expression, but only weak activation of DUX4 target genes and no widespread apoptosis (*Ganassi et al., 2022*). After induction, myoblasts were washed and supplemented with fresh medium without doxycycline. Samples were harvested for RNA extraction in triplicate immediately after washing and then at 1, 2, 3, 4, 5, 6, 8, and 10 hr post-wash. RT-qPCR was performed employing a standard curve to quantify the DUX4 mRNA copy number over our time course, to monitor DUX4 mRNA degradation in the absence of induction (*Figure 2A*).

As anticipated DUX4 mRNA levels decayed exponentially over time in the absence of doxycycline (linear regression of ln(DUX4 mRNA) vs time, p=4.1 × 10⁻⁴, *Figure 2B*), allowing us to calculate the degradation rate of DUX4 mRNA, $d_0 = 0.246$/hr. This estimate suggests that the half-life of DUX4 mRNA is approximately 2.8 hr, not atypical for a transcription factor, and on the faster end of the mRNA degradation distribution (*Yang et al., 2003*).

To estimate the mean transcription rate of DUX4, $V_D$, we applied the promoter-switching model presented above. We considered the scRNAseq dataset of FSHD single myocytes produced by *van den Heuvel et al., 2019*. This dataset comprises 7047 single myocytes, which were differentiated for 3 days in the presence of EGTA to inhibit fusion. 5133/7047 (73%) single myocytes were derived from four FSHD patients (two FSHD1 and two FSHD2). Patient demographics, genotypes, and DUX4 +/-ve cell counts are displayed in *Supplementary file 1*. DUX4 mRNA expression was detected in 27/5133 (0.53%) single FSHD myocytes but not in control myocytes (*van den Heuvel et al., 2019*; *Banerji and Zammit, 2019*). Considering the 5133 FSHD myocytes together we implemented a gradient descent algorithm to approximate the MLEs of the normalized parameters $k_a^{DUX4}/\delta^{DUX4}$, $k_i^{DUX4}/\delta^{DUX4}$ and $v_0^{DUX4}/\delta^{DUX4}$ from the Poisson-Beta interpretation of the promoter-switching model of DUX4 expression (*Figure 2C*).

We note that $\delta^{DUX4} = d_0$, which we have already estimated, this enabled us to renormalize these parameters and compute the average DUX4 transcription rate, $V_D := \frac{k_a^{DUX4} v_0^{DUX4}}{k_a^{DUX4} + k_i^{DUX4}} = 0.00211$/hr.

As the total number of DUX4 positive cells is low, we pooled data from the four FSHD patients to allow robust estimation of the average DUX4 transcription rate for this patient cohort. We note, however, that distinct FSHD genotypes likely underlie different DUX4 transcription rates. The majority of DUX4 +ve cells (23/27) were found in two FSHD patients for this scRNAseq data. For patient FSHD1.1 19/2226 (0.85%) cells were DUX4 +ve and for patient FSHD2.1 5/1283 (0.39%) cells were

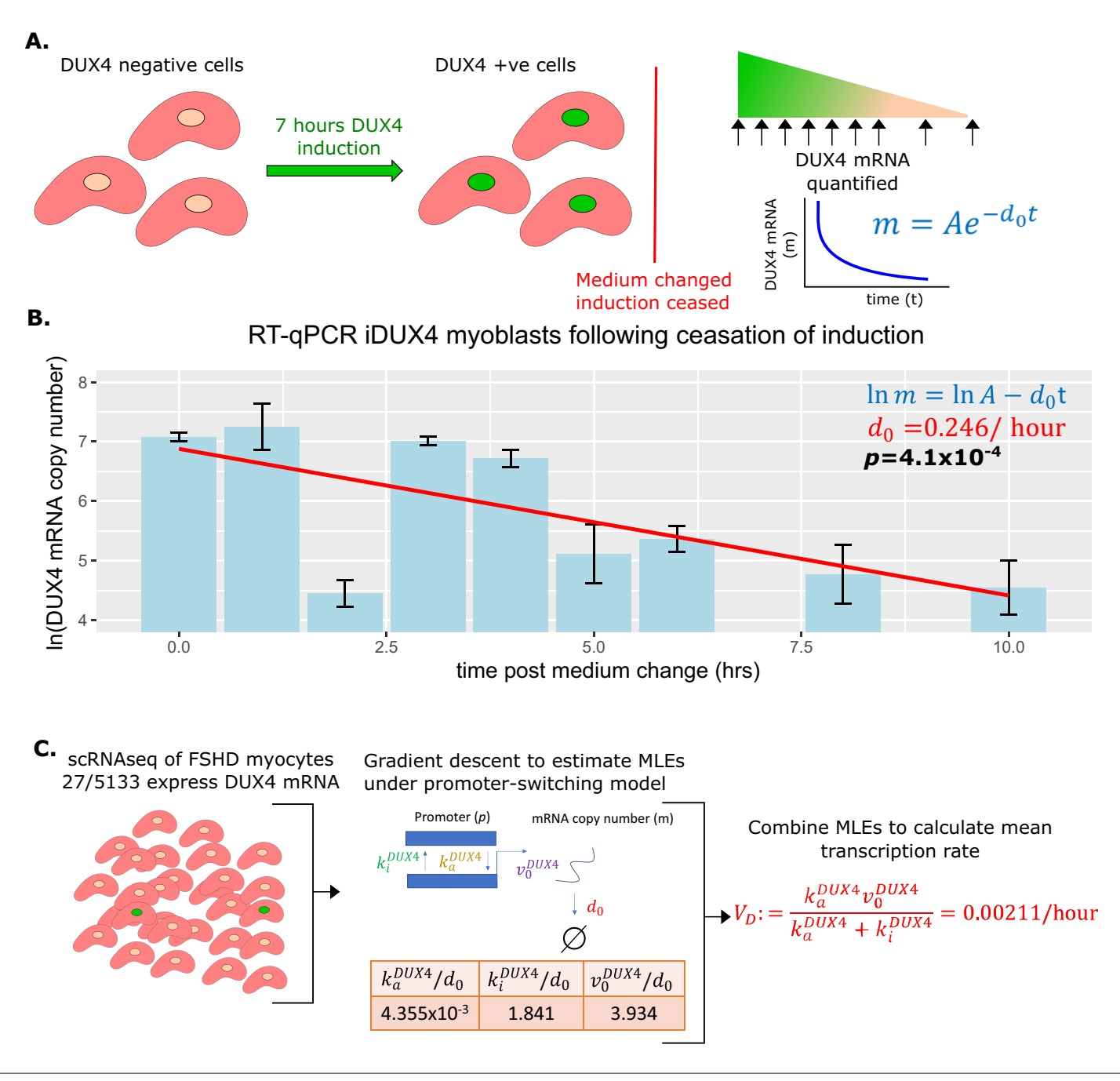

**Figure 2.** Estimation of DUX4 mRNA degradation rate $(d_0)$ and average transcription rate $(V_D)$. (**A**) Schematic of the experiment performed to estimate the DUX4 mRNA degradation rate $d_0$. iDUX4 myoblasts were induced to express DUX4 with 250 ng/ml of doxycycline for 7 hr before doxycycline was washed away. DUX4 mRNA was quantified at multiple time points post-wash using RT-qPCR. (**B**) Bar chart displays the RT-qPCR of ln(DUX4 copy number) at post-wash times 0, 1, 2, 3, 4, 5, 6, 8, and 10 hr. Bar represents the average of triplicates and the standard error of the mean is displayed. A line of best fit of ln(DUX4 copy number) against time is displayed alongside the corresponding linear regression $p$-value (bold) and the slope of the line corresponds to $d_0$ (red). (**C**) Schematic of the estimation of average DUX4 transcription rate $V_D$ from scRNAseq data of 5133 Facioscapulohumeral muscular dystrophy (FSHD) myocytes. The maximum likelihood estimates (MLEs) for the underlying parameters of DUX4 transcription under the promoter-switching model are estimated via gradient descent and combined to estimate the average DUX4 transcription rate (red text).

DUX4 +ve (*Supplementary file 1*). To investigate variability in $V_D$, we derived individual estimates for these two 'higher' DUX4 expressing patients, for FSHD1.1 $V_D = 0.00373$/hr, while for FSHD2.1 $V_D = 0.000960$/hr. The pooled estimate of $V_D$ is thus comparable in order of magnitude with that of individual patients. To fully utilize the available data and prevent limiting our model to only 'higher' DUX4 expressing patients, we employ the pooled estimate of $V_D = 0.00211$/hr for the remainder of our calculations.

## Estimating kinetics of DUX4 target activation

We next estimated the average transcription rate of the DUX4 target genes in the presence of DUX4 mRNA, $V_T$. We focused on eight DUX4 target genes: *ZSCAN4, TRIM43, RFPL1, RFPL2, RFPL4B, PRAMEF1, PRAMEF2,* and *PRAMEF12* that have been identified as direct DUX4 targets via ChIP-seq (*Young et al., 2013*). We have shown that these eight genes are the only features consistently up-regulated in human myoblasts expressing DUX4, across four independent studies (*Banerji and Zammit, 2021*; *Rickard et al., 2015*; *Choi et al., 2016*; *Young et al., 2013*; *Geng et al., 2012*).

Examining mRNA levels of these eight DUX4 target genes in scRNAseq (*van den Heuvel et al., 2019*) and snRNAseq (*Jiang et al., 2020*) studies of FSHD and control differentiated myoblasts, expression was restricted to FSHD cells/nuclei and never observed in controls. This pattern of expression mirrors that of DUX4 mRNA and suggests that these targets are highly specific and are unlikely to be activated in the absence of DUX4.

To confirm our hypothesis, we applied the promoter-switching model. We returned to the scRNAseq data of 5133 FSHD myocytes generated by *van den Heuvel et al., 2019* and divided the data into the 27 DUX4 +ve cells and 5106 DUX4 -ve cells. For each of our eight DUX4 target genes, we implemented our gradient descent algorithm to compute the MLEs of the normalized parameters $k_a^n/\delta^n$, $k_i^n/\delta^n$, and $v_0^n/\delta^n$, underlying a promoter-switching model for the given gene across the 27 DUX4 +ve FSHD myocytes and the 5106 DUX4 -ve FSHD myocytes separately (*Figure 3A*).

In the presence of DUX4 mRNA, the proportion of time the promoters of the eight DUX4 target genes remained in the active state, $\frac{k_a^n}{k_a^n + k_i^n}$, significantly increased (paired Wilcoxon p=7.8 × $10^{-3}$, *Figure 3B*), as expected. Curiously, however, the normalized rate of transcription of the DUX4 target genes from the active promoter, $v_0^n/\delta^n$, significantly decreased in the presence of DUX4 (paired Wilcoxon p=0.039, *Figure 3C*).

To investigate further, we considered the moments of the distribution of mRNA copy number, $m$, under the promoter-switching model. It can be shown that the mean mRNA copy number satisfies (*Vu et al., 2016*):

$$E\left[m\right] = \frac{k_a^n v_0^n / \delta^n}{k_a^n + k_i^n}.$$

The changes in parameters we calculate for the DUX4 target genes confirmed that in the presence of DUX4 mRNA, the mean expression of all 8 DUX4 targets increases (paired Wilcoxon p=0.039, *Figure 3D*), i.e., the drop in $v_0^n/\delta^n$ is over-compensated for by the rise in $\frac{k_a^n}{k_a^n + k_i^n}$. However, it is curious that $v_0^n/\delta^n$ should drop at all.

It can be shown that the variance of mRNA copy number $m$, satisfies (*Vu et al., 2016*):

$$Var\left[m\right] = E\left[m\right]\left(1 + \frac{k_i^n v_0^n / (\delta^n)\hat{2}}{(1 + k_a^n/\delta^n + k_i^n/\delta^n)(k_a^n/\delta^n + k_i^n/\delta^n)}\right).$$

This expression ensures that as the mean mRNA copy number rises, so too must the variance, however, the level to which the variance rises is controlled by a term that is monotonic increasing in $v_0^n/\delta^n$ and depends on the promoter state parameters in a more complex way.

We postulated the pattern of DUX4 target gene parameter changes we observe in the presence of DUX4, have the effect of raising mean target mRNA expression, while suppressing target mRNA variance, i.e., a controlled activation of target genes. To investigate this, we considered two hypothetical scenarios in which the mean expression of target mRNAs, $E\left[m\right]$ in the absence of DUX4, is increased to the same level we observe in the presence of DUX4. In the first scenario, we considered only increasing $v_0^n / \delta^n$ to achieve the rise in $E\left[m\right]$, in the second we considered only increasing the ratio of active to inactive promotor $\frac{k_a^n}{k_i^n}$. Both hypothetical scenarios resulted in a significantly higher variance of

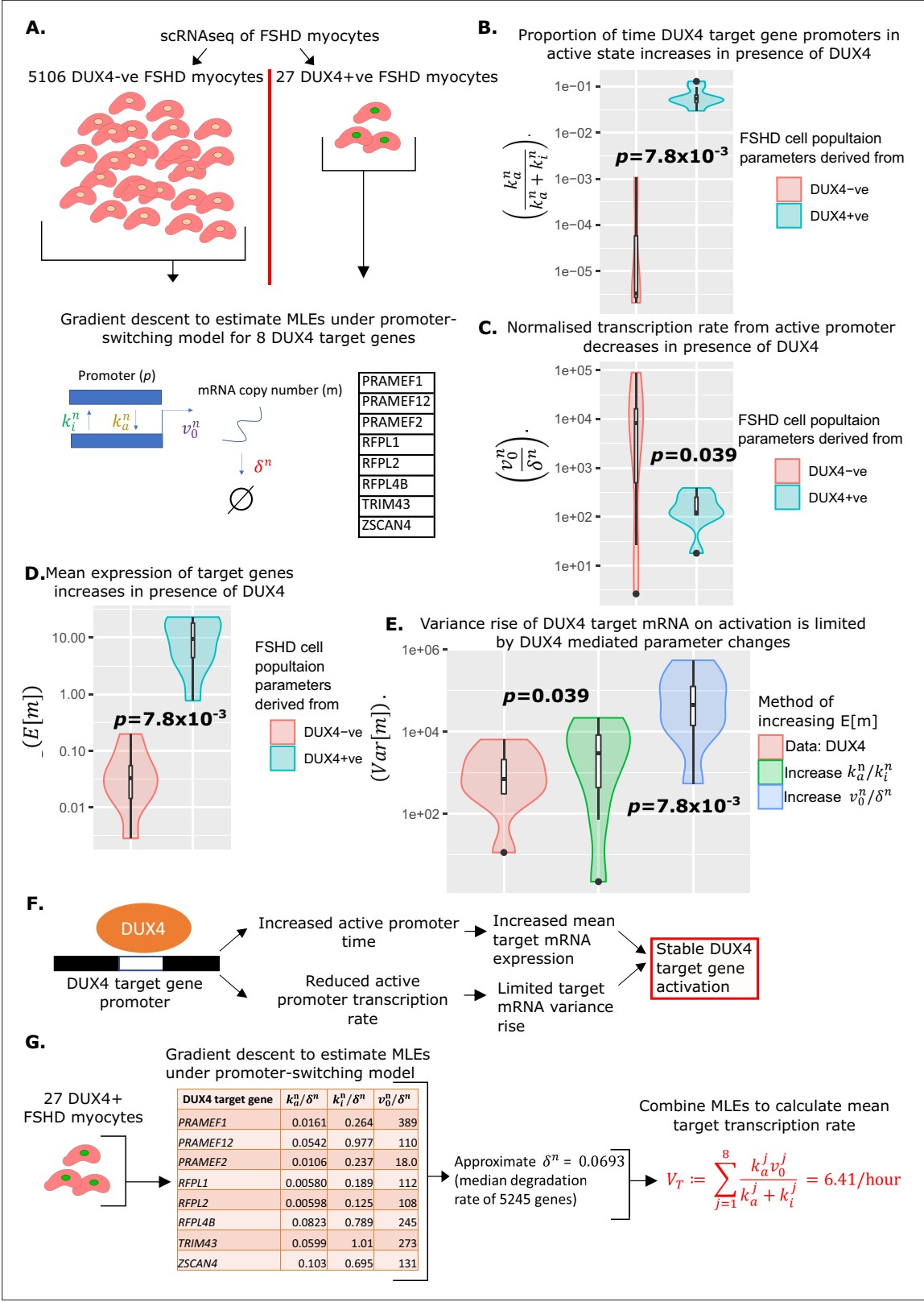

**Figure 3.** Estimation of promoter-switching model parameters for DUX4 target genes and estimation of $V_T$. (**A**) Schematic of estimation of promoter-switching model parameters for eight DUX4 target genes, across 5106 DUX4 -ve Facioscapulohumeral muscular dystrophy (FSHD) single myocytes and 27 DUX4 +ve FSHD single myocytes from **van den Heuvel et al., 2019**. Violin plots display (**B**) the proportion of time in active promoter state (**C**) normalized active promoter transcription rate and (**D**) the mean mRNA copy number, for the eight DUX4 target genes in the 5106 DUX4 -ve and

*Figure 3 continued on next page*

*Figure 3 continued*

27 DUX4 +ve FSHD myocytes separately, *p*-values correspond to two-tailed paired Wilcoxon tests. (**E**) Violin plot displays the variance of mRNA copy number for the eight DUX4 target genes calculated from the 27 DUX4 +ve FSHD myocytes (red) and calculated assuming DUX4 up-regulates targets only via the increase in (blue) the normalized transcription rate $\frac{v_0^n}{\delta^n}$ or (green) the ratio of active to inactive promoter transition rates $\frac{k_a^n}{k_i^n}$. Paired two-tailed Wilcoxon *p*-values are displayed comparing adjacent distributions. (**F**) Schematic displaying how the change in parameters underlying the promoter-switching models for the eight DUX4 target genes in the presence of DUX4 leads to stable target gene activation. (**G**) Schematic of the estimation of the average DUX4 target gene transcription rate $V_T$, from scRNAseq data of 27 DUX4 +ve FSHD myocytes.

The online version of this article includes the following source data for figure 3:

**Source data 1.** Source data for *Figure 3* provides the parameters of the promoter switching model for each of the 8 DUX4 targets, derived from 5106 DUX4 -ve FSHD single cells and 27 DUX4 +vs FSHD single cells seperately.

---

target mRNA than observed in our data, with the pure rise in $v_0^n/\delta^n$ driving the most dramatic increase in target mRNA variance (*Figure 3E*).

Taken together these results suggest that under our promoter-switching model, increasing the expression of a gene comes at the cost of increasing the variance of its expression, and that the greatest contributor to this variance comes from the normalized active promoter transcription rate $v_0^n / \delta^n$ . The parameter changes we observe in DUX4 target genes, suggest a management of this trade-off by DUX4, which increases the mean expression of target mRNA through a large increase in the proportion of time the promoter is active, $\frac{k_a^n}{k_a^n+k_i^n}$, while offsetting the resulting rise in variance through a modest decrease in normalized active promotor transcription $v_0^n/\delta^n$ (*Figure 3F*).

We formulate the mean transcription rate of at least 1 of the 8 DUX4 targets from our compartment model, $V_T$, as the sum of the mean transcription rates of all eight target genes in the presence of DUX4 mRNA, i.e.,:

$$V_T \sum_{j=1}^{8} \frac{k_a^j v_0^j}{k_a^j + k_i^j}$$

where $j$ indexes the eight DUX4 target genes, and the promoter-switching model parameters are estimated from the 27 DUX4 expressing FSHD single myocytes. Our promoter-switching model scRNAseq-derived MLEs are normalized parameters $k_a^n/\delta^n$, $k_i^n/\delta^n$ and $v_0^n/\delta^n$. We must, therefore, estimate $\delta^n$ for each target gene to compute $V_T$ . As there is a range of target genes, we approximate $\delta^n$ for each by the median mRNA degradation rate observed in an analysis of 5245 genes (*Yang et al., 2003*), and set $\delta^n = 0.0693$ resulting in $V_T = 6.41$/hr (*Figure 3G*).

As with our calculation of $V_D$ data was pooled across four FSHD patients to calculate $V_T$ . We do not anticipate patient genotype to impact the average DUX4 target transcription rate, independently of its impact on the DUX4 transcription rate. However, to confirm our findings on the impact of DUX4 on target gene promoter dynamics, in a patient-specific setting, we attempted calculation of the parameters underlying the promoter switching model for the eight DUX4 target genes in DUX4 +ve and DUX4-ve cells, for patients FSHD1.1 and FSHD2.1 separately. Due to the limited number of cells for each patient, personalised estimates for all eight target genes could not be obtained. However, where patient-specific estimates have obtained the direction of parameter differences in target genes, between DUX4 +ve and DUX4 -ve cells were in line with those of pooled estimates across four FSHD patients (*Supplementary file 2*).

The remaining two parameters of our compartment model were calculated from published data. For the translation rate of DUX4 mRNA to active protein $T_D$ , we considered our analysis of iDUX4 myoblasts (*Ganassi et al., 2022*). We induced DUX4 expression with 250 ng/ml doxycycline and performed RT-qPCR to assess the expression of *DUX4*, and 3/8 of our DUX4 target genes *ZSCAN4*, *TRIM43*, and *PRAMEF1*, at 7 hr, 16 hr, and 24 hr of induction. DUX4 mRNA levels peaked at 7 hr, while the expression of DUX4 target genes peaked between 16 and 24 hr (*Ganassi et al., 2022*). This suggests a delay between DUX4 mRNA production and the presence of active DUX4 protein of between 9 and 17 hr, on average 13 hr. We thus estimate $T_D = \frac{1}{13}$/hr.

For the death rate of DUX4 target positive cells, $D_r$ , we consider the data of *Rickard et al., 2015*, in which differentiating FSHD myoblasts containing a DUX4-activated GFP reporter were imaged every 15 min for 120 hr. Following activation of the DUX4 reporter, cells died ~20.2 hr later (*Rickard et al., 2015*). We thus estimate $D_r = \frac{1}{20.2}$/hr.

## Compartment model simulation

Having defined experimental estimates for parameters underlying the compartment model, we simulated the model forward in time to observe how an initial distribution of cells progresses through the five compartments.

To provide a ground truth we considered the scRNAseq data of *van den Heuvel et al., 2019*. Defining a DUX4 target gene +ve cell as expressing at least one of our eight DUX4 target genes, we assign the 5133 FSHD myocytes to 1 of the 4 live cell states of our compartment model: $S\,(3\,days) = 4956\,\,(96.6\%)$, $E(3\,days) = 14(0.273\%)$, $I\,(3\,days) = 13\,\,(0.253\%)$, $R\,(3\,days) = 150\,\,(2.92\%)$.

If we assume that at the start of differentiation, all FSHD myocytes occupy state $S\,(0)$ being DUX4 negative and DUX4 target gene negative, simulating our model we estimated that 7% of starting cells will have died over 3 days. To replicate the starting conditions of the scRNAseq data we thus set $S\,(0) = 5133\,(1 + 0.07) = 5488$, $E\,(0) = I\,(0) = R\,(0) = D\,(0) = 0$. Simulating our model over 3 days from

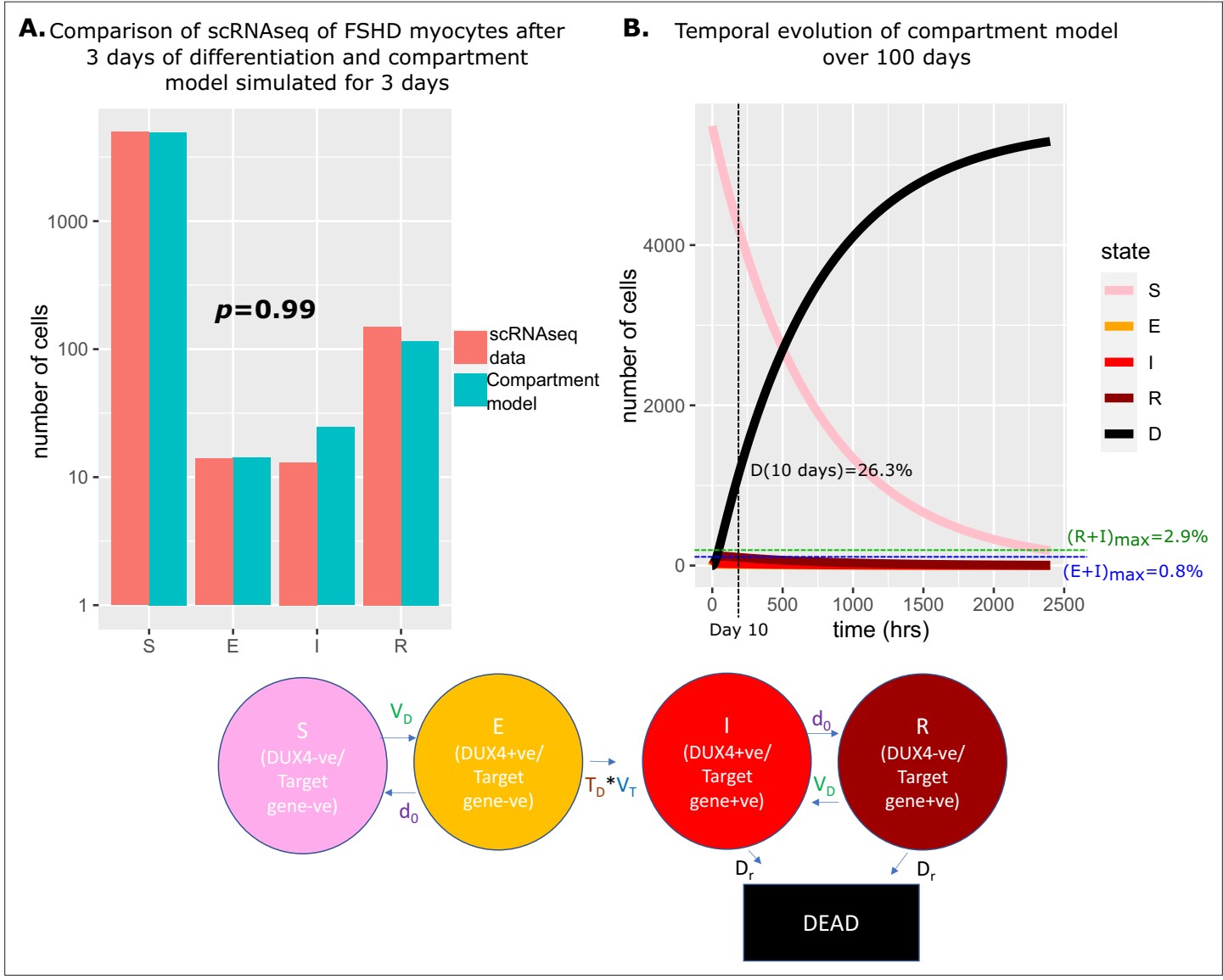

**Figure 4.** Simulation of the compartment model and comparison with scRNAseq data of unfused Facioscapulohumeral muscular dystrophy (FSHD) myocytes. (**A**) Bar chart displays the number of cells in each live cell compartment (blue) in our model following simulation over 3 days using experimentally estimated parameters and (red) in 5133 single FSHD myocytes. A Chi-squared goodness of fit *p*-value tests the alternate hypothesis that the two distributions are different. (**B**) Line plot displaying how the number of cells in each of the five compartments of the model changes over 100 days from a starting state of 5133 (1+0.07) cells. The percentage of cells dead after 10 days is displayed alongside the maximum percentage of live cells which are DUX4 +ve $(E+I)_{max}$ and DUX4 target gene +ve $(R+I)_{max}$.

this starting condition, we obtained cell state proportions statistically indistinguishable from the experimental scRNAseq data: $S\left(3\,days\right) = 4953\ \left(96.5\%\right), E\left(3\,days\right) = 14(0.253\%),\ I\left(3\,days\right) = 25\ \left(0.487\%\right),$ $R\left(3\,days\right) = 116\ \left(2.26\%\right)$ (Chi-Squared p=0.99, *Figure 4A*).

We next simulated our model forward over 100 days to observe how the proportion of cells in each state changed (*Figure 4B*). As expected, the number of cells in the DUX4 naive, susceptible state $S\left(t\right)$ gradually decreased, while the number of dead cells due to DUX4 $D\left(t\right)$ gradually rose, so that after 10 days 26.3% of cells had died as a consequence of DUX4 expression. Remarkably despite cell death in our model only being attributable to DUX4 expression, the dynamics predict that this is achieved while keeping the number of DUX4 mRNA and DUX4 target mRNA positive cells extremely low. DUX4 positive cells never rose to more than 0.8% of the live cell population and DUX4 target positive cells never more than 2.9%.

Our compartment model with experimentally derived parameters thus provides an excellent fit to real-world data and gives a mechanism for how extremely low levels of DUX4 and target gene expression can drive significant cell death in FSHD myocytes.

## Modeling DUX4 expression in FSHD myotubes allows syncytial diffusion

Having considered scRNAseq data of unfused FSHD myocytes differentiated for 3 days, in which DUX4 expressed in a given nucleus cannot interact with other nuclei, we next considered syncytial diffusion of DUX4. *Jiang et al., 2020* published a single myonuclear RNAseq (snRNAseq) of FSHD2 myoblasts differentiated over 3 days to form multinucleated myotubes. The data describes 139 FSHD2 myonuclei and 76 control myonuclei. As with the scRNAseq data *van den Heuvel et al., 2019*, none of the control myonuclei express DUX4 nor any of the 8 DUX4 target genes. For FSHD myonuclei, 3/139 (2.2%) express DUX4. The number of myonuclei in each of the live 'cell' compartments of our model is: $S\left(3\,days\right) = 58\ \left(41.7\%\right), E(3\,days) = 0(0\%), I\left(3\,days\right) = 3\ \left(2.2\%\right),$ $R\left(3\,days\right) = 78\ \left(56.1\%\right)$.

Comparing the proportions of myonuclei (cells) assigned to the four live states of our compartment model in the unfused FSHD scRNAseq and syncytial FSHD snRNAseq datasets, we found significant differences in the distributions (Chi-Squared p<2.2 × 10⁻¹⁶, *Figure 5A*). The unfused FSHD myocytes demonstrated a higher proportion of DUX4 -ve, DUX4 target gene -ve *susceptible (S)* cells, while the syncytial FSHD myonuclei demonstrated a greater proportion of DUX4 -ve, DUX4 target gene +ve *resigned (R)* cells.

We investigated whether allowing diffusion of DUX4 protein between myonuclei in our model could explain this difference in proportions. Two states in our compartment model express DUX4 target genes and thus have evidence for DUX4 protein: $I\left(t\right)$ and $R\left(t\right)$, while states $S\left(t\right)$, and $E\left(t\right)$ do not. We updated our model to the syncytial compartment model, where we allow the DUX4 protein compatible states to 'infect' the DUX4 protein incompatible states at a rate Δ (*Figure 5B and C*).

The term 'infect' is only an analogy and our model is not based on a direct mapping of e.g., viral infection to our setting. In our case, the 'infectious agent' DUX4 causes harm but does not replicate. DUX4 protein is a transcription factor and thus activates the expression of target genes. In our model, DUX4 can be seen as an 'infectious agent', and expression of DUX4 target genes can be seen as the 'infection', which leads to cell death. DUX4 can either stay in the 'infected' cell until it dies or diffuse to another cell to spread the 'infection'.

We next employed a genetic algorithm to fit our syncytial compartment model to the snRNAseq data (*Jiang et al., 2020*). As differentiating FSHD myoblasts take approximately 24 hr to initiate fusion (*Banerji et al., 2019*), and DUX4 expressing cells show defects in fusion (*Knopp et al., 2016*), we assumed that day 3 of differentiation represents day 2 of syncytial myonuclei and simulated our model over 48 hr, from a starting state of $S\left(0\right) = n$, $E\left(0\right) = I\left(0\right) = R\left(0\right) = 0$. We optimized two parameters via our genetic algorithm: the DUX4 syncytial diffusion rate Δ, and the number of starting myonuclei in the susceptible state $n$ (*Figure 5D*).

The algorithm converged on a solution of $n = 217$ susceptible myonuclei and Δ = 7.46 × 10⁻⁴ / hr. Simulating the syncytial compartment model over 2 days employing these parameters resulted in a statistically indistinguishable approximation to the snRNAseq data (*Jiang et al., 2020*): $S\left(3\,days\right) = 62\ \left(44.6\%\right), E\left(3\,days\right) = 0(0\%),\quad I\left(3\,days\right) = 1\ \left(0.720\%\right),\quad R\left(3\,days\right) = 76\ (54.7\%),$ (Chi-Squared p=0.99, *Figure 5E*). Thus differences in proportions of cell states in fused versus unfused

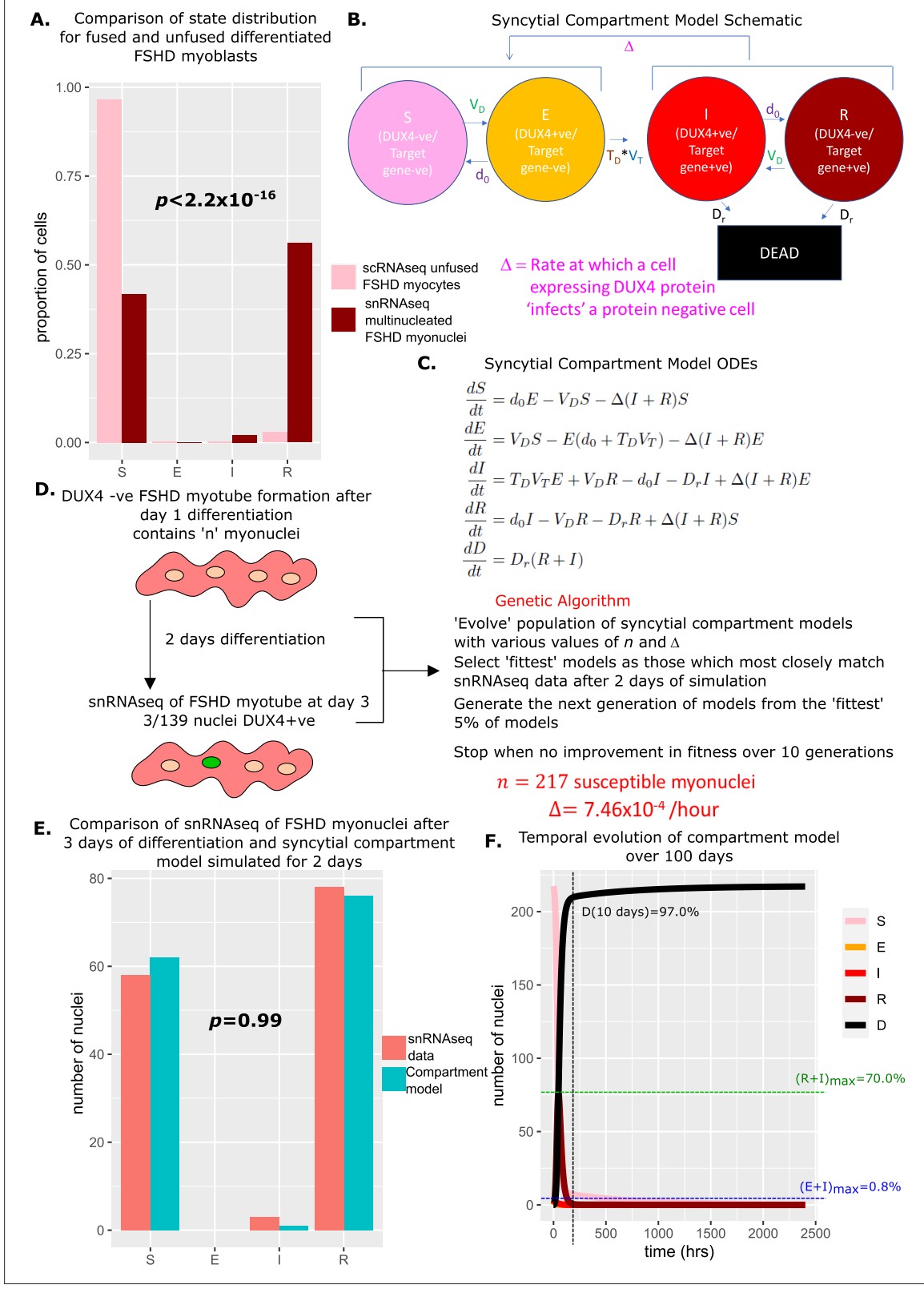

**A.** Comparison of state distribution for fused and unfused differentiated FSHD myoblasts

$p<2.2\times10^{-16}$

scRNAseq unfused FSHD myocytes

snRNAseq multinucleated FSHD myonuclei

**B.** Syncytial Compartment Model Schematic

$\Delta$

S (DUX4-ve/Target gene-ve) — $V_D$ → E (DUX4+ve/Target gene-ve) — $T_D*V_T$ → I (DUX4+ve/Target gene+ve) — $d_0$ → R (DUX4-ve/Target gene+ve)

$d_0$   $V_D$

$D_r$   $D_r$

DEAD

$\Delta$ = Rate at which a cell expressing DUX4 protein 'infects' a protein negative cell

**C.** Syncytial Compartment Model ODEs

$$\frac{dS}{dt} = d_0 E - V_D S - \Delta(I+R)S$$
$$\frac{dE}{dt} = V_D S - E(d_0 + T_D V_T) - \Delta(I+R)E$$
$$\frac{dI}{dt} = T_D V_T E + V_D R - d_0 I - D_r I + \Delta(I+R)E$$
$$\frac{dR}{dt} = d_0 I - V_D R - D_r R + \Delta(I+R)S$$
$$\frac{dD}{dt} = D_r(R+I)$$

**Genetic Algorithm**

'Evolve' population of syncytial compartment models with various values of $n$ and $\Delta$

Select 'fittest' models as those which most closely match snRNAseq data after 2 days of simulation

Generate the next generation of models from the 'fittest' 5% of models

Stop when no improvement in fitness over 10 generations

$n = 217$ susceptible myonuclei
$\Delta = 7.46\times10^{-4}$ /hour

**D.** DUX4 -ve FSHD myotube formation after day 1 differentiation contains 'n' myonuclei

2 days differentiation

snRNAseq of FSHD myotube at day 3
3/139 nuclei DUX4+ve

**E.** Comparison of snRNAseq of FSHD myonuclei after 3 days of differentiation and syncytial compartment model simulated for 2 days

$p=0.99$

snRNAseq data

Compartment model

**F.** Temporal evolution of compartment model over 100 days

D(10 days)=97.0%

$(R+I)_{max}$=70.0%

$(E+I)_{max}$=0.8%

S, E, I, R, D

**Figure 5.** Derivation and simulation of the syncytial compartment model and comparison with snRNAseq data of fused Facioscapulohumeral muscular dystrophy (FSHD) myonuclei. (**A**) Bar chart displays the proportion of cells in each live compartment of our model in (pink) 5133 unfused FSHD single myocytes from **van den Heuvel et al., 2019** and (dark red) 139 fused FSHD single myonuclei of **Jiang et al., 2020**. A Chi-squared goodness of fit $p$-value tests the alternate hypothesis that the two distributions are different. (**B**) Schematic of the syncytial compartment model, allowing DUX4

*Figure 5 continued on next page*

*Figure 5 continued*

protein permissive states *I* and *R*, to 'infect' non-permissive states *S* and *E*. (**C**) Ordinary differential equations describing the syncytial compartment model. (**D**) Schematic of the genetic algorithm employed to estimate the DUX4 diffusion rate Δ and the starting population size *n*, for the syncytial compartment model from 139 fused FSHD single myonuclei. (**E**) Bar chart displays the number of cells in each live cell compartment (blue) in our syncytial compartment model following simulation over 2 days using experimentally estimated parameters and (red) in 139 fused single FSHD myonuclei. A Chi-squared goodness of fit *p*-value tests the alternate hypothesis that the two distributions are different. (**F**) Line plot displaying how the number of cells in each of the five compartments of the syncytial compartment model changes over 100 days from a starting state of *n* cells. The percentage of cells dead after 10 days is displayed alongside the maximum percentage of live cells which are DUX4 +ve (E+I)$_{max}$ and DUX4 target gene +ve (R+I)$_{max}$.

myocytes can be explained by the addition of a diffusion term for DUX4 protein, within the limits of the assumptions of our model.

Simulating the syncytial compartment model over 100 days, we found a much faster death rate than the unfused myocyte model, with 97% of myonuclei dead by day 10. DUX4 mRNA-expressing cells remain low in proportion, never exceeding 0.8% of total live nuclei, however, DUX4 target gene mRNA expressing cells can now comprise a significant proportion of living nuclei, up to 70% at 73 hr, though on average comprise only 5.6% of living nuclei (*Figure 5F*).

## A cellular automaton model of FSHD myotubes

A limitation of our syncytial compartment model is the assumption that any DUX4 protein-positive myonucleus can 'infect' any DUX4 protein-negative myonucleus. In practice, DUX4 can likely only spread between adjacent myonuclei in a short-range interaction (*Rickard et al., 2015*; *Tassin et al., 2013*). To overcome this limitation, we re-cast our compartment model as a cellular automaton.

In this grid-based model, squares on the grid represent myonuclei, by introducing an appropriate boundary condition, the grid is topologically equivalent to the surface of a cylinder and thus can be considered to represent myonuclei residing on the surface of a myofiber (*Figure 6A*).

We evolved our cellular automaton forwards in time in two steps. First, the non-syncytial compartment model was applied to each myonucleus to update the internal state of each nucleus stochastically according to the experimentally derived transition rates and the time period that has elapsed. In the second step, myonuclei in DUX4 protein-compatible states $I(t)$ and $R(t)$ can 'infect' any of the eight neighbouring myonuclei in the grid which are in states $S(t)$ or $E(t)$ at a stochastic rate Δ (*Figure 6B*). Our cellular automaton model, thus retains all features of the compartment model, but facilitates a more realistic topology for a muscle fiber and prevents DUX4 protein from engaging in long-range non-physiological interactions.

We next employed a genetic algorithm to fit the cellular automaton to the syncytial snRNAseq data (*Jiang et al., 2020*). We again assumed that day 3 of differentiation corresponds to day 2 of syncytial myotubes and simulate our model over 48 hr, in time steps of 1 hr, from a starting state of $S(0) = n$, $E(0) = I(0) = R(0) = 0$. However, we now optimized three parameters: the DUX4 local diffusion rate Δ, the myotube nuclear length $l$ and the myotube nuclear circumference $c$, where $n = lc$.

As the cellular automaton model is stochastic due to the discrete time steps employed, one cannot expect the same set of parameters to yield the same distribution of cell states after 48 hr of simulation. Thus the solution of a genetic algorithm may represent a parameter regime optimal for replicating the snRNAseq data, or may represent an unlikely realization of a parameter regime sub-optimal for replicating the data. To overcome this limitation we implemented 100 genetic algorithms to obtain a 'family' of parameter values Δ and $c$, which underlie potentially optimal solutions (*Figure 6C*).

We performed a clustering analysis of the 100 optimal solutions, employing the parameter values Δ, $l$ and $c$ as a feature space. This revealed that a cluster of 14 solutions has significantly greater myotube length than circumference (Wilcoxon p=5.0 × 10⁻⁸). This implies that for these 14 solutions, the genetic algorithm has converged to the general structure of a muscle fiber cylinder (long and thin). We focused on the mean parameter values of this cluster as the most physiologically realistic solution: Δ=0.0402/hr , $l = 44$ myonuclei, and $c = 9$ myonuclei (*Figure 6D*).

Simulating the cellular automaton model 100 times with this parameter range over 48 hr resulted on average in a statistically indistinguishable approximation of the snRNAseq syncytial FSHD myonuclei state distribution (Chi-squared p=0.98, *Figure 6E*).

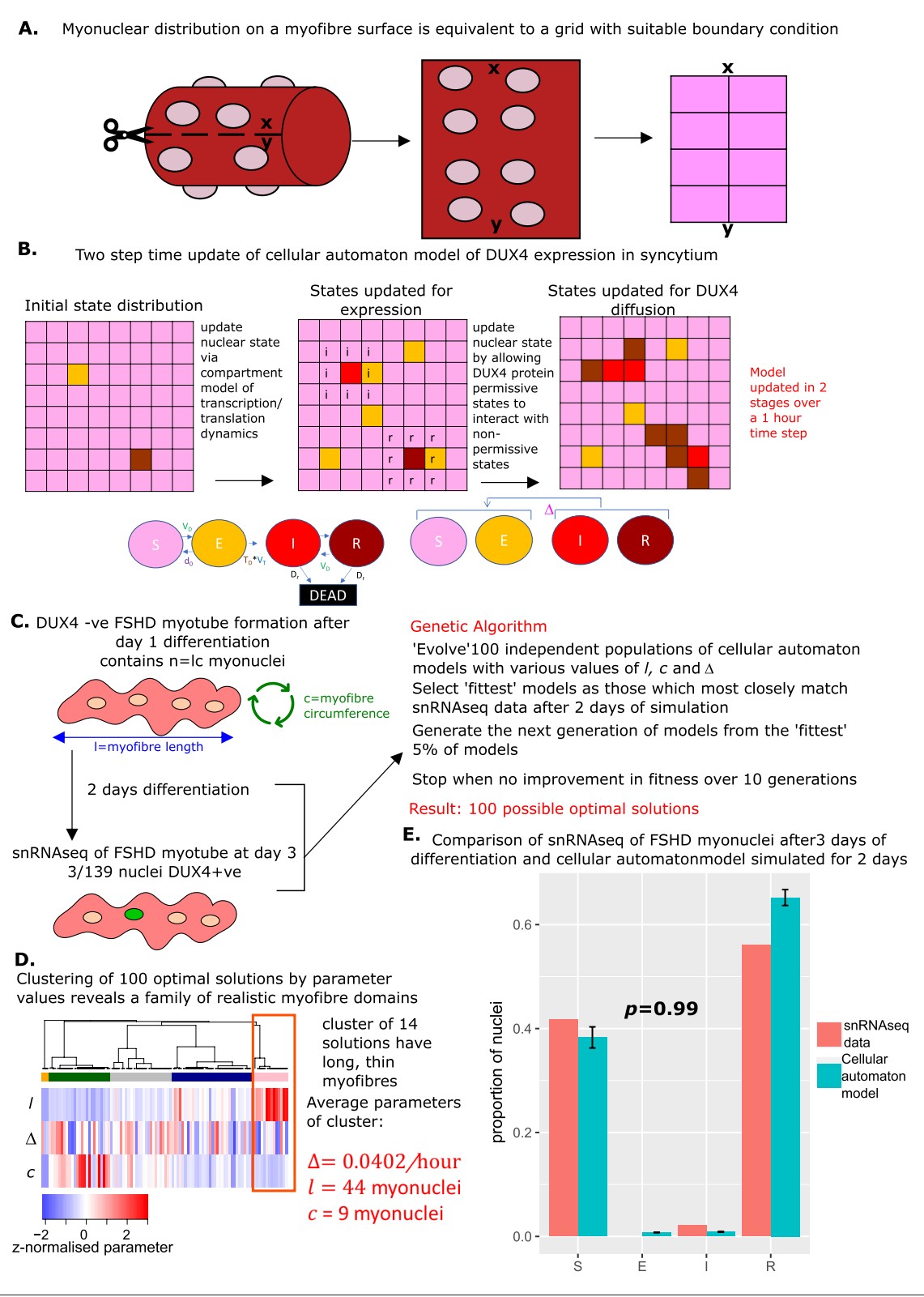

**Figure 6.** Derivation and simulation of the cellular automaton model and comparison with snRNAseq data of fused Facioscapulohumeral muscular dystrophy (FSHD) myonuclei. (**A**) Schematic demonstrating how with suitable boundary conditions a grid can be topologically equivalent to a cylinder, and thus encapsulate dynamics on the surface of a myofiber. (**B**) Schematic of the cellular automaton model of DUX4 expression in a syncytium with grid squares representing single myonuclei updating in 1 hr time-steps via a two-stage process. (**C**) Schematic of the genetic algorithm employed to

*Figure 6 continued on next page*

*Figure 6 continued*

estimate the DUX4 diffusion rate $\Delta$, myotube length $l$, and myotube circumference $c$ for the cellular automaton model from 139 fused FSHD single myonuclei. (**D**) Heatmap displays the clustering solution of the 100 optimal parameter regimes $l, c$ and $\Delta$ produced by fitting 100 genetic algorithms to 139 fused FSHD myonuclei. The highlighted cluster represents a parameter regime in which myotubes are long and thin, the average parameter values of this cluster are displayed in red. (**E**) Bar chart displays the average proportion of cells in each live cell compartment (blue) in our cellular automaton model following simulation over 2 days using experimentally estimated parameters, alongside standard error of the mean, and (red) in 139 fused single FSHD myonuclei. A Chi-squared goodness of fit *p*-value tests the alternate hypothesis that the two distributions are different.

Thus after restricting DUX4 protein from spreading to only neighboring nuclei, we were still able to explain the difference between unfused and fused cell state distributions purely with a diffusion term for DUX4 protein.

## A toolbox for assessing the impact of therapies on DUX4-mediated myotoxicity

Anti-DUX4 therapy for FSHD can target any aspect of DUX4 expression. To understand the impact of a given therapeutic strategy one requires two key pieces of information:

1. The true value of the parameters underlying DUX4 expression.
2. How a proposed change in these parameters by a therapeutic will impact cell death.

Here, we have derived experimental estimates for the parameters underlying DUX4 expression which can be modified by anti-DUX4 therapy, while our compartmental and cellular automaton models provide the framework for investigating how parameter changes will impact cell death.

To facilitate other investigators using our models to guide anti-DUX4 therapy development, we have packaged them into graphical user interfaces to provide three user-friendly tools.

The first tool allows investigators to visualize the compartment model and syncytial version, for a range of parameter choices. Sliders allow the investigator to perturb the six parameters of our models: $V_D$ , $d_0$, $V_T$, $T_D$, $D_r$, and $\Delta$, and outputs the temporal evolution of the models over as many hours as required, from any starting number of *susceptible* cells/nuclei, as well as histograms comparing the perturbed model to the experimental data of *van den Heuvel et al., 2019* and *Jiang et al., 2020*.

The second tool implements the cellular automaton model of DUX4 in a syncytium. As with the first tool users can employ sliders to perturb the 6 parameters of the model $V_D$ , $d_0$, $V_T$, $T_D$, $D_r$, and $\Delta$, as well as the dimensions of the myofiber grid $l$ and $c$, and the number of hours over which the automaton should be simulated. The automaton can then be started and the nuclear state updates will be dynamically played on a grid, while a histogram dynamically updates the proportions of nuclei in each state.

The final tool allows investigators to compare dead cell proportions over time for a chosen parameter regime (($V_D$) , $d_0$, $V_T$, $T_D$, $D_r$, and $\Delta$, $l$ and $c$), with cell death under our experimentally derived parameters. Users can again select their parameter regime and dynamically view changes in dead cell proportions under the single cell compartment model, to simulate realizations of the cellular automaton model and perform comparisons of cell death via Cox proportional hazard models.

The tools are hosted at the following web addresses:

1. Compartment Models: https://crsbanerji.shinyapps.io/compartment_models/
2. Cellular Automaton: https://crsbanerji.shinyapps.io/ca_shiny/
3. Survival Analysis: https://crsbanerji.shinyapps.io/survival_sim/

The three tools can be used to understand how anti-DUX4 therapies aimed at different model parameters alter the proportion of cells in each compartment of our model at given times. This can guide the optimal therapeutic approaches, as well as optimal time points to assay to validate given therapies using cell culture approaches.

## Discussion

Anti-DUX4 therapy is the leading candidate for an FSHD treatment, with several compounds currently in clinical trials (*Tawil, 2020*; *Jagannathan et al., 2022*; *Le Gall et al., 2020*). However, DUX4 expression in FSHD muscle demonstrates a complex dynamic (*Banerji and Zammit, 2021*; *Banerji and*

*Zammit, 2019*). Understanding this complex dynamic is essential to the construction of optimal therapy, as it is currently unclear which stage of the DUX4 'central-dogma' one should target to have the most significant impact on pathology. This has led to diverse strategies, targeting epigenetic regulation of DUX4 (*Lemmers et al., 2012*; *Block et al., 2013*), DUX4 mRNA (*Wallace et al., 2012*; *Ciszewski et al., 2020*), DUX4 protein (*Klingler et al., 2020*), and DUX4 downstream effects (*Heher et al., 2022*).

Here, we present a mathematical model of DUX4 expression in differentiated FSHD myoblasts, based on ordinary differential equations and stochastic gene expression. By analyzing human myoblasts expressing inducible DUX4 as well as scRNAseq and snRNAseq of FSHD patient myocytes and myotubes, we compute experimental estimates for the parameters underlying our model. These include the first estimates of DUX4 transcription, translation, and mRNA degradation rates. Simulating our model with experimentally derived parameters we find that it accurately predicts the proportion of DUX4 +ve/-ve and DUX4 target gene +ve/-ve cells observed in actual scRNAseq of FSHD patient myocytes. We package our model into graphical user interface tools to allow investigators to rapidly observe the impact of any given anti-DUX4 therapy on cell viability.

As with all models, ours are subject to assumptions, such as combining the multi-step processes of transcription and promoter activation into single steps. We further assumed that cells expressing our 8 DUX4 target genes are historic DUX4-expressing cells. We selected these eight genes as they are confirmed DUX4 targets by ChIP-seq and have been identified as upregulated in every transcriptomic analysis of DUX4 over-expressing human myoblasts (*Banerji and Zammit, 2021*). Tissue expression patterns of these eight genes are not well characterized outside of the FSHD context, however, their expression has been reported during zygotic genome activation (*Taubenschmid-Stowers et al., 2022*) and in testicular tissue (*Taubenschmid-Stowers et al., 2022*; *Ishiguro et al., 2017*), both settings where DUX4 is physiologically expressed. We found further evidence for the expression of these genes indicating historic DUX4 in this study. First, we never find a single transcript for any of these eight target genes in 1914 single myocytes and 77 single nuclei from control individuals, suggesting that their expression requires an FSHD genotype (and thus likely DUX4 expression). Second, our investigation of these genes in scRNAseq data of FSHD myocytes demonstrated that in the presence of DUX4, the expression of all eight genes significantly increases, indicating activation by DUX4. While this is evidence that the eight targets are specific to DUX4, it is possible that they may also be induced at a much lower level, by some other factor related to the FSHD genotype and thus a small proportion of DUX4 target gene +ve cells in this study may not be historically DUX4 +ve.

We also limited our model to differentiated muscle culture, when DUX4 expression is more robust. DUX4 mRNA is typically detected in a small proportion (0.5–3.8%) of FSHD myoblasts in vitro (*Jiang et al., 2020*; *van den Heuvel et al., 2019*; *Banerji and Zammit, 2019*), and protein expression is similarly rare (0.1%) (*Snider et al., 2010*). The situation is qualitatively different in FSHD muscle biopsies, where DUX4 mRNA detection is highly variable and DUX4 protein detection is extremely rare. DUX4 target gene expression is detected in inflamed FSHD muscle biopsies but less robustly in non-inflamed muscle, implying that a systemic component may also activate DUX4 expression in vivo (*Banerji and Zammit, 2021*; *Banerji, 2020*; *Banerji et al., 2020b*). Despite the added complexity of DUX4 expression in the muscle biopsy setting, all anti-DUX4 therapies currently under consideration were first investigated in myogenic cells in vitro. Thus, it is encouraging that we can simulate DUX4 and target gene expression single-cell distributions which are indistinguishable from those observed in FSHD patient muscle cell culture. Through theoretical investigation, we can also explore processes such as DUX4 protein syncytial diffusion and DUX4 target gene activation, which may be less accessible to experimentation.

FSHD is a rare disease and public datasets describing scRNAseq and snRNAseq of patient-derived primary myocytes are currently limited to those used in this study. Consequently, we have pooled data describing 4 individual patients of different FSHD genotypes to estimate two of the parameters of our model: $V_T$ and $V_D$ . DUX4 expression levels depend on D4Z4 repeat length and methylation status, and can differ between cell lines isolated from different FSHD patients (*Jones et al., 2012*; *Homma et al., 2012*), as well as between genetically identical cell lines isolated from the same mosaic FSHD patient (*Krom et al., 2012*). We computed patient-specific estimates of $V_D$ for 2/4 patients with sufficient data and found these comparable to the pooled estimate. By pooling patients, we obtained parameter estimates that may not be true for an individual but represented the 'average' parameter

values across these four FSHD patients. Our in silico model of DUX4 expression is proposed as a pre- in vitro screening tool to guide anti-DUX4 therapy for the general population. However, as more data on FSHD is generated our models will evolve. In particular, as higher volumes of scRNAseq data describing FSHD patients with different genotypes become available, the model can be updated to facilitate genotype-stratified and even personalized anti-DUX4 therapy design. Moreover, new data describing scRNAseq of proliferating FSHD primary myoblasts, alongside quantification of how DUX4 and DUX4 target gene expression impacts myoblast proliferation rates, would allow natural extension of our model to understand DUX4 expression during FSHD myoblast proliferation.

DUX4 is myotoxic (*Kowaljow et al., 2007*), but how its rare expression drives and sustains significant pathology is unclear. DUX4 has been proposed to undergo a rare burst-like expression dynamic (*Bosnakovski et al., 2017*; *Tassin et al., 2013*), though no attempt has previously been made to understand DUX4 expression as a stochastic process. Here, we model DUX4 and its target gene expression directly as stochastic processes via our promoter-switching model and estimate the underlying parameters. Our cell-biology informed model predicts that in single FSHD myocytes/myotubes, DUX4 can drive significant cell death, eventually depleting the entire population, while being expressed in only 0.8% of live cells. This level matches well with the proportion of DUX4-positive cells seen in published studies (*Banerji and Zammit, 2021*; *van den Heuvel et al., 2019*; *Snider et al., 2010*) and confirms that a burst-like mechanism of DUX4 expression can drive significant pathology while making DUX4 difficult to detect.

DUX4 has also been proposed to spread through the myofiber syncytium from its originator nucleus to 'infect' DUX4 naïve nuclei, bypassing the need for a rare expression burst and accelerating pathology (*Tassin et al., 2013*; *Barro et al., 2010*). The proportion of DUX4 target positive cells is greater in snRNAseq of syncytial FSHD myotubes than scRNAseq of unfused myocytes, while the proportion of DUX4 positive cells is comparable, supporting a syncytial diffusion mechanism. We model DUX4 in FSHD myotubes as an infectious agent, able to spread from one nucleus to another (but not replicate) by adapting epidemiological compartment models, which we package as a cellular automaton to provide a realistic myotube surface on which to monitor DUX4 expression. Incorporation of a diffusion term is sufficient to account for the difference in DUX4 target gene +ve/-ve nuclear proportions, between syncytial FSHD myotubes and unfused FSHD myocytes, while maintaining comparable proportions of DUX4 +ve cells. We thus demonstrate that the theoretical hypothesis of DUX4 syncytial diffusion is compatible with biological data, and provide an estimate for the DUX4 diffusion rate.

Our model predicts <2.9% of single FSHD myocytes will be DUX4 target gene mRNA positive at any given time, despite significant DUX4-driven cell death. While in FSHD myotube nuclei this proportion can transiently spike at 70%, on average it remains much lower at ~5.6%. This suggests that in FSHD muscle, expression of DUX4 target genes at any given time typically remains restricted to a small number of myonuclei. This may explain why DUX4 target gene expression is an inconsistent biomarker of FSHD muscle (*Banerji et al., 2017*; *Banerji and Zammit, 2019*). Moreover, our model predicts that DUX4 target gene expression does not typically increase over time, supporting the finding that DUX4 target gene expression in FSHD muscle has not been validated as a marker of disease progression (*Banerji, 2020*; *Wong et al., 2020*). Lastly, due to the typically low levels observed, it may be difficult to detect changes in DUX4 target gene activation following anti-DUX4 therapies. This may have contributed to the recent phase 2b trial of losmapimod in FSHD failing to reach its primary outcome measure of suppressing DUX4 target gene expression in patient muscle biopsies (*Jagannathan et al., 2022*).

By modeling DUX4 target gene expression as stochastic processes, we found that DUX4 increases the proportion of time target gene promoters are in an active configuration, while curiously decreasing the normalized active promoter transcription rate. Through analytic investigation of our model, we found the rise in active promotor time drove increased DUX4 target gene expression, while the drop in normalized active promotor transcription rate dampened the noise associated with this higher expression level. This suggests that DUX4 modifies the expression of its target genes to orchestrate a precise, stable activation. It has been shown that DUX4 is a pioneer factor and increases target gene promoter activation via C-terminal recruitment of p300 to drive H3K27 acetylation (*Choi et al., 2016*). How DUX4 may limit normalized active promoter transcription rates of its target genes is unclear, but could involve interaction with the transcriptional complex, or feeding back to decrease the stability of target mRNA.

We thus present our model of DUX4 expression as a theoretical setting to understand the complex dynamics of this important disease gene and as an open source, in silico platform to rapidly and cheaply pre-screen anti-DUX4 therapy for FSHD.

# Methods

## Key resources table

| Reagent type (species) or resource | Designation | Source or reference | Identifiers | Additional information |
|---|---|---|---|---|
| Cell line (human) | LHCN-M2-iDUX4 | 10.1038/s41598-018-35150-8 | | Immortalized doxycycline-inducible DUX4 expressing human myoblast cell line |
| Sequence-based reagent | DUX4 Fwd | 10.1038/s41598-018-35150-8 | qPCR pimer | ACCTCTCCTAGAAACGGAGGC |
| Sequence-based reagent | DUX4 Rev | 10.1038/s41598-018-35150-8 | qPCR primer | CAGCAGAGCCCGGTATTCTTC |
| Commercial assay or kit | Quantitect reverse transcription kit | Qiagen, 205311 | RT-kit | |
| Commercial assay or kit | Takyon SYBR Green qPCR Mastermix | UF-NSMT-B101 | qPCR Mastermix | |

## Cell culture

DUX4 inducible immortalized human myoblasts LHCN-M2 were cultured in Promocell growth media (C-23060) supplemented with 15% FBS and 1:1000 Gentamycin (Sigma), in a humidified incubator at 5% $CO_2$. Cells were found negative for mycoplasma contamination (MycoAlert PLUS Mycoplasma, Lonza).

## RT-qPCR

iDUX4 myoblasts were induced to express DUX4 with 250 ng/ml doxycycline for 7 hr in six-well plates, before washing with PBS and the addition of fresh medium without doxycycline. Total mRNA was isolated using the RNeasy Kit (Qiagen, 74–104) according to the manufacturer's instructions in triplicate immediately after washing and then at 1, 2, 3, 4, 5, 6, 8, and 10 hr post-wash. After reverse transcription with the Quantitect reverse transcription kit (Qiagen, 205311), SYBR green qPCR was performed (Takyon, UF-NSMT-B101) using a Viia7machine (ThermoFisher). DUX4 primers were: Fwd ACCTCTCCTAGAAACGGAGGC, Rev CAGCAGAGCCCGGTATTCTTC.

## Standard curve to compute $d_0$

The length of the DUX4 construct used in our standard curve was 8938 bp and the number of copies per ng was calculated via the following formula:

$$copy\ number/ng\ :\ =\ \frac{N_A}{8938\left(660 \times 10^{-9}\right)} = 1.02 \times 10^9$$

Where $N_A = 6.022 \times 10^{23}$ mol$^{-1}$ is Avogadro's number and 660 g/mol is the molecular weight of a single DNA base pair. Linear regression of the standard curve confirmed that DUX4 threshold cycles (Cts) are a linear function of $\log_{10}\left(copy\ number\right)$ ($R^2$=0.994), with a slope of 30.4 and intercept of –2.70. These values were employed to convert DUX4 Cts from our iDUX4 assay to DUX4 copy number via the following formula:

$$copy\ number = 10^{\frac{Ct - 30.4}{-2.70}}$$

In our iDUX4 assay, we sample RNA immediately after washing off doxycycline and at multiple time points post-wash. We assume that no DUX4 can be transcribed post-wash and model the DUX4 copy number as an exponential decay:

$$DUX4\ copy\ number = Ae^{-\,d_0 t};$$

$$\ln\left(DUX4\ copy\ number\right) = \ln A -\ d_0 t$$

Linear regression of $\ln\left(DUX4\ copy\ number\right)$ against time post wash ($t$) yielded a slope of 0.246, giving our estimate of $d_0$, the degradation rate of DUX4.

## scRNAseq and snRNAseq data

Normalized read counts for scRNAseq of 7047 FSHD and controls single myocytes produced by **van den Heuvel et al., 2019** were downloaded from the GEO data base accession GSE122873. This dataset contains 5133 single myocytes derived from four FSHD patients (two FSHD1 and two FSHD2) and 1914 single myocytes from two control individuals. Myocytes were differentiated for 3 days in the presence of EGTA to prevent fusion.

Normalized read counts for snRNAseq of 317 FSHD and controls single myonuclei produced by **Jiang et al., 2020** were downloaded from the GEO database accession GSE143492. This data comprised 47 FSHD2 nuclei at day 0 of differentiation and 139 FSHD2 nuclei at day 3, from two seperate patients, alongside 54 control nuclei at day 0 and 77 control nuclei at day 3 from two separate individuals. For comparison with the scRNAseq data, we focused on myonuclei from day 3 of differentiation.

All myocytes and myonuclei nuclei were assigned to 1 of 4 live cell compartments via the following criteria:

1. *S* – no detectable normalized reads for *DUX4, ZSCAN4, TRIM43, RFPL1, RFPL2, RFPL4B, PRAMEF1, PRAMEF2,* and *PRAMEF12*.
2. *E* – at least one detectable normalized *DUX4* read, and no detectible normalized reads for *ZSCAN4, TRIM43, RFPL1, RFPL2, RFPL4B, PRAMEF1, PRAMEF2,* and *PRAMEF12*.
3. *I* – at least one detectable normalized *DUX4* read, and at least one detectible normalized read for any of *ZSCAN4, TRIM43, RFPL1, RFPL2, RFPL4B, PRAMEF1, PRAMEF2,* and *PRAMEF12*.
4. *R* – no detectable normalized reads for *DUX4*, and at least one detectible normalized read for any of *ZSCAN4, TRIM43, RFPL1, RFPL2, RFPL4B, PRAMEF1, PRAMEF2,* and *PRAMEF12*.

All control myocytes/myonuclei were in the *S* compartment. The distribution of the four live cell compartments in the 5133 FSHD single myocytes and the 139 days three FSHD single myonuclei were compared via a Chi-squared test, and significance was assessed at the 5% level.

## Estimation of MLEs of promoter-switching model parameters from scRNAseq data

To calculate the transcription rate for DUX4 and DUX4 targets we adopt the approach employed by Larsson et al. and Trung et al. where a maximum likelihood estimation (MLE) methodology is used to infer the parameters for the two-state model of stochastic gene expression (**Vu et al., 2016**; **Larsson et al., 2019**). This is implemented in the *poisbeta* python package available at https://github.com/aksarkar/poisbeta. The procedure takes the Poisson-Beta distribution discussed in the results:

$$p \sim \text{Beta}\left(k_a^n/\delta^n, k_i^n/\delta^n\right)$$

$$m|p \sim \text{Poisson}\left(pv_0^n/\delta^n\right)$$

and minimizes the negative log-likelihood for each gene

$$\arg_{\min\left(\theta\ \text{in}\ \Theta\right)} \sum_{i=1}^{N} -\ln\text{P}\left(m_i|\theta\right),$$

where $p$ is a Beta-distributed variable describing the promotor state, $N$ is the number of cells, $m_i$ is the number of mRNA transcripts for cell $i = 1, \ldots, N$, and $\theta = \left[v_0^n/\delta^n, k_a^n/\delta^n, k_i^n/\delta^n\right]$, is the parameter space. From here on we will not include the normalizing $\delta^n$ term for brevity.

As a first estimate of $\theta$, we use the method of moments derived by Peccoud and Ycart for the two-state promotor model (**Peccoud and Ycart, 1995**)

$$v_0^n = \frac{-r_1 r_2 + 2r_1 r_3 - r_2 r_3}{r_1 - 2r_2 + r_3},$$

$$k_a^n = \frac{2\left(r_2 - r_1\right)\left(r_1 - r_3\right)\left(r_3 - r_2\right)}{\left(r_1 r_2 - 2r_1 r_3 + r_2 r_3\right)\left(r_1 - 2r_2 + r_3\right)},$$

$$k_i^n = \frac{2r_1\left(r_3 - r_2\right)}{r_1 r_2 - 2r_1 r_3 + r_2 r_3},$$

where

$$r_1 = e_1,$$

$$r_2 = \frac{e_2}{e_1},$$

$$r_3 = \frac{e_3}{e_2},$$

and

$$e_1 = \frac{1}{N} \sum_{i=1}^{N} m_i,$$

$$e_2 = \frac{1}{N} \sum_{i=1}^{N} m_i (m_i - 1),$$

$$e_3 = \frac{1}{N} \sum_{i=1}^{N} m_i (m_i - 1)(m_i - 2).$$

$\theta$ is then optimized via the Nelder-Mead method using the Gauss-Jacobi quadrature to evaluate the likelihood function

$$\mathfrak{l} = \frac{1}{\text{Beta}\left(k_a^n, k_i^n\right)} \int_0^1 \text{Poisson}\left(m_i; v_0^n p_i\right) p_i^{k_a^n - 1} \left(1 - p_i\right)^{k_i^n - 1} dp_i$$

$$\mathfrak{l} = \frac{1}{2^{k_a^n + k_i^n - 1} \text{Beta}\left(k_a^n, k_i^n\right)} \int_{-1}^1 \text{Poisson}\left(m_i; v_0^n \frac{1 + t_i}{2}\right) (1 + t_i)^{k_a^n - 1} \left(1 - p_i\right)^{k_i^n - 1} dt_i$$

$$\mathfrak{l} \approx \frac{1}{2^{k_a^n + k_i^n - 1} \text{Beta}\left(k_a^n, k_i^n\right)} \sum_{k=1}^{K} w_k \text{Poisson}\left(m_i; v_0^n y_k\right)$$

where $p_i = (t_i + 1)/2$ to substitute the beta-distributed parameter, $K$ is the number of points integrated over, $w_k$ is the weight of the Jacobi polynomial of degree $k$, and $y_k$ is the root. The optimizer, quadrature method, beta distribution, and Poisson distribution are implemented in the scipy package available from https://scipy.org/install/.

## Statistical comparison of promoter-switching model parameters

The three normalized parameters of the promoter-switching model $v_0^n/\delta^n, k_a^n/\delta^n, \ k_i^n/\delta^n$ , were computed as above for each of the 8 DUX4 target genes *ZSCAN4, TRIM43, RFPL1, RFPL2, RFPL4B, PRAMEF1, PRAMEF2*, and *PRAMEF12*, in the 27/5133 FSHD single myocytes with detectable DUX4 mRNA and separately in the 5106 FSHD single myocytes without detectable DUX4, in the dataset of **van den Heuvel et al., 2019**.

The normalized active promoter transcription rate $v_0^n/\delta^n$ , the proportion of time spent with a promoter active $\frac{k_a^n}{k_a^n + k_i^n}$ and the mean mRNA expression level $E[m]$ for the eight DUX4 target genes were compared between DUX4 positive and DUX4 negative myocytes via paired two-tailed Wilcoxon tests, significance was assessed at the 5% level.

## Hypothetical scenarios for raising $E[m]$ under the promoter-switching model

We considered two hypothetical scenarios for raising $E[m]$ under the promotor-switching model to observe the impact on $Var[m]$ . In the first $\frac{v_0^n}{\delta^n}$ is increased while keeping other parameters constant. In the second $\frac{k_a^n}{k_i^n}$ is increased keeping other parameters constant. Simple algebraic manipulation allows analytical solutions for the raised parameters and thus $Var[m]$ .

In the case of $v_0^n/\delta^n$ , we assume that $\frac{v_0^n}{\delta^n} \to x \frac{v_0^n}{\delta^n}$ for some $x > 1$, the rise in $E[m]$ in this case is simply:

$$E\left[m|x\frac{v_0^n}{\delta^n}\right] - E\left[m|\frac{v_0^n}{\delta^n}\right] = \frac{\frac{k_a^n v_0^n}{\delta^n}}{k_a^n + k_i^n}(x - 1)$$

We can thus choose $x$, to achieve the rise in $E[m]$ seen under DUX4 expression $d(\mu)$ as:

$$x = 1 + d\left(\mu\right) \frac{k_a^n + k_i^n}{\frac{k_a^n v_0^n}{\delta^n}}$$

Which can be substituted into the formula for $Var\left[m\right]$ to compute $Var\left[m | x \frac{v_0^n}{\delta^n}\right]$, giving the expected variance of the mRNA copy distribution if $E\left[m\right]$ is increased by $d\left(\mu\right)$, solely by increasing $\frac{v_0^n}{\delta^n} \rightarrow x \frac{v_0^n}{\delta^n}$ for some $x > 1$.

In the case of $\frac{k_a^n}{k_i^n}$, we assume that $\frac{k_a^n}{k_i^n} \rightarrow x \frac{k_a^n}{k_i^n}$, we further assume that the rise in active transition rate is mirrored by a drop in the inactivation rate, i.e., the two parameters are not mutually exclusive, thus: $k_a^n \rightarrow y k_a^n$ and $k_i^n \rightarrow \frac{1}{y} k_i^n$, where $y^2 = x$. The rise in $E\left[m\right]$ in this case is:

$$E\left[m | x \frac{k_a^n}{k_i^n}\right] - E\left[m | \frac{k_a^n}{k_i^n}\right] = \frac{v_0^n k_a^n}{\delta^n}\left(\frac{x}{x k_a^n + k_i^n} - \frac{1}{k_a^n + k_i^n}\right)$$

We can thus choose $x$, to achieve the rise in $E\left[m\right]$ seen under DUX4 expression $d\left(\mu\right)$ as:

$$x = \frac{k_i^n\left(E\left[m | \frac{k_a^n}{k_i^n}\right] + d\left(\mu\right)\right)}{k_i^n E\left[m | \frac{k_a^n}{k_i^n}\right] + d\left(\mu\right) k_a^n}$$

Which can be substituted into the formula for $Var\left[m\right]$ to compute $Var\left[m | x \frac{k_a^n}{k_i^n}\right]$.

For each of the 8 DUX4 target genes we considered the parameters of the promoter switching model $v_0^n/\delta^n, k_a^n/\delta^n, k_i^n/\delta^n$, computed over the 5106 DUX4 -ve FSHD single myocytes from the dataset of van den Heuvel et al., as the starting parameters to input into the above formulae for $x$. Comparing the mean expression of each target in the 27 DUX4 +ve myocytes and the 5106 DUX4 -ve myocytes we computed $d\left(\mu\right)$. For each target we thus computed $Var\left[m | x \frac{v_0^n}{\delta^n}\right]$ and $Var\left[m | x \frac{k_a^n}{k_i^n}\right]$, the target mRNA copy number variances under our two hypothetical scenarios. These variances were compared with the true mRNA copy number variance in the 27 DUX4 +ve myocytes, as well as with each other via two-tailed paired Wilcoxon tests, significance was assessed at the 5% level.

## Compartment model simulation

After the estimation of the five parameters $V_D$, $d_0$, $V_T$, $T_D$, and $D_r$, the ODEs underlying the compartment model (*Figure 1B*) were simulated from an initial state of all cells in state $S\left(0\right)$, over a period of 100 days in timesteps of 1 hr using the deSolve package in R (*Soetaert et al., 2010*). For comparison with the 5133 single FSHD myocytes differentiated for 3 days described by *van den Heuvel et al., 2019*, an initial population sized 5133 was simulated for 3 days, after which 6.901% of starting cells were in the dead cell state. In order to replicate the live cell population of 5133 after 3 days we thus employed a starting population of 5133 (1+0.06901) cells in state $S\left(0\right)$. After simulation for 3 days the proportion of cells in each live cell state $S\left(3\, days\right)$, $E\left(3\, days\right)$, $I\left(3\, days\right)$, and $R\left(3\, days\right)$ was compared between the model simulation and the true scRNAseq data, via a Chi-squared test.

For the syncytial compartment model, simulation was performed as above with $V_D$, $d_0$, $V_T$, $T_D$, and $D_r$ unchanged. The additional parameter $\Delta$ was estimated via the genetic algorithm (below) alongside the starting population $n$ of cells in state $S$. After simulation for 2 days (as there is assumed no syncytium for the first 24 hr of fusion) the proportion of cells in each live cell state was compared between the model simulation and the true snRNAseq data of *Jiang et al., 2020* via a Chi-Squared test. Significance was assessed at the 5% level.

## Cellular automaton model

The cellular automaton model was evolved on an $l \times c$ rectangular grid describing the length and circumference of the myotube. Individual cells on the grid correspond to single myonuclei in a syncytium. Each cell was evolved stochastically over a discrete time-step of 1 hr. In each time-step 2 independent actions were performed.

First, the state of each cell was evolved according to the non-syncytial compartment model. Transition from connected states was modeled as an exponential process with a rate parameter equal to the transition rate (e.g. transition from $S$ to $E$ occurred according to $\exp\left(V_D\right)$). An event occurring or not under this exponential distribution within 1 hr was simulated during each time step. If the event occurred a transition in cell state occurred, otherwise the cell state remained the same. In the situation

where a cell state could transition in two possible directions (e.g. state $E$ can transition to $S$ at a rate $d_0$ and to $I$ at a rate $T_D V_T$) competing exponential clocks were set up. Using the property that $\min\left(\exp\left(A\right), \exp\left(B\right)\right) \sim \exp\left(A + B\right)$, we simulated a transition occurring in either direction within 1 hr. If a transition occurred, we simulated the exponential clocks describing transition in either direction independently, and transitioned the cell state according to whichever experienced an event faster. If no transition occurred the cell state remained the same.

Once every cell was evolved according to the non-syncytial compartment model we updated the model according to the diffusion of DUX4. Cells in states $I$ or $R$ at the start of the time-step can interact with cells in states $S$ or $E$ if they are among their eight immediate neighbors (*Figure 6B*). Each interaction was again modeled as an exponential distribution with rate $\Delta$, which was simulated for an event occurring within 1 hr. If the event occurred then the state of the neighboring cell was updated $S \rightarrow I$ or $E \rightarrow R$, otherwise the state of the neighboring cell remained the same.

We implement a boundary condition so that cells at position 1 on the circumference dimension are able to interact with cells in position $c$, to allow our grid topological equivalence to a cylinder (*Figure 6A*).

The cellular automaton model was evolved from a starting distribution of all cells in state $S$ over 48 hr time-steps, employing the parameters $V_D$, $d_0$, $V_T$, $T_D$ and $D_r$ estimated from the non-syncytial compartment model and $\Delta$, $l$ and $c$ estimated from the genetic algorithm and clustering analysis (see below). Due to the stochastic nature of the exponential clocks involved, 100 simulations were performed to obtain an average behavior of cell type proportions under our model. The proportion of cells in each of the live cell states obtained from the average behavior of our model was compared to the true snRNAseq data of *Jiang et al., 2020* via a Chi-Squared test. Significance was assessed at the 5% level.

## Genetic algorithms

We employed genetic algorithms twice, firstly in the syncytial compartment model to fit the number of myonuclei $n$ in the starting state $S$ and the diffusion parameter $\Delta$. Second in the cellular automaton model to again fit $\Delta$, and the length $l$ and circumference $c$ of the automaton grid, and thus again the number of cells $n = l \times c$ in the starting state $S$.

Both algorithms employed the same fitness function aimed at minimizing the difference in the proportion of cells in the live states between the simulated model after 2 days and the snRNAseq of *Jiang et al., 2020*:

$$-\left( \left| \frac{\frac{S^{sim}\left(2days\right)}{n^{live}}}{\frac{59}{139}} + \frac{\frac{E^{sim}\left(2days\right)}{n^{live}}}{\frac{\varepsilon}{139}} + \frac{\frac{I^{sim}\left(2days\right)}{n^{live}}}{\frac{3}{139}} + \frac{\frac{R^{sim}\left(2days\right)}{n^{live}}}{\frac{78}{139}} \right| \right)$$

Where $n^{live} n - D^{sim}\left(2days\right)$ and $\varepsilon \ll 1$ is chosen to prevent singularity while rewarding models which obtain values of $\frac{E^{sim}\left(2days\right)}{n^{live}}$ close to 0. This fitness function was chosen as it outperformed conventional functions based on Minkowski and Euclidean distances. A value of $\varepsilon = 0.1$, proved sufficient to generate models with live cell state proportions indistinguishable from the snRNAseq data via the Chi-squared test.

Genetic algorithms were implemented using the GA package in R (*Scrucca, 2013*) using default parameters: starting population of 50 models, elitism of 0.05, mutation probability 0.1, crossover probability 0.8, and an optimal model was selected if maximum fitness was stable for 10 generations.

The cellular automaton model evolves stochastically and thus an optimal fitness function for a population could represent an optimal parameter regime, or an unlikely simulation of a sub-optimal parameter regime. We thus ran 100 genetic algorithms on the cellular automaton to generate 100 optimal parameter regimes and examined their structure via clustering analysis (below).

## Clustering analysis

Clustering analysis was performed on the $3 \times 100$ feature space of parameter regimes $\Delta$, $l$ and $c$ found optimal in the 100 genetic algorithms performed on the cellular automaton model. The Consensus-ClusterPlus package in R (*Wilkerson and Hayes, 2010*) implemented K-medoids clustering using a Euclidean distance metric, and consensus-CDF cluster stability plots ascertained the optimal number of clusters in the parameter feature space as 5.

Two-tailed paired Wilcoxon tests comparing the distribution of $l$ and $c$ values in each cluster, demonstrated that only one parameter cluster output significantly long and thin myotubes, in line with the microanatomy of muscle (*Figure 6D*). The average values of $\Delta$, $l$, and $c$ from this cluster were thus considered the optimal parameter regime for the cellular automaton model.

## Shiny tools

Three GUI tools were written using the shiny package in R (*Chang et al., 2014*). The first tool outputs the compartment models for user parameter inputs and is implemented using the deSolve package (*Soetaert et al., 2010*). The second tool outputs a dynamic realization of the stochastic cellular automaton model for user parameter inputs, the automaton is displayed using the lattice package in R (*Sarkar & Deepayan, 2008*). The third tool displays the proportion of dead cells under the compartment model comparing our experimentally derived parameter regime to a user-selected parameter regime. In addition, the third tool simulates two realizations of the cellular automaton model, one using our experimentally derived parameter regime and another using the user-selected regime ($l$ and $c$ are kept the same for both simulations to allow comparable starting populations). Survival analysis using Cox-proportional hazard models is implemented via the survival package in R (*Therneau and Grambsch, 2018*), to compare cell death rates in the two realizations, with *p*-values displayed on a corresponding Kaplan-Meier plot.

The three tools can be accessed at:

1. Compartment Models: https://crsbanerji.shinyapps.io/compartment_models/
2. Cellular Automaton: https://crsbanerji.shinyapps.io/ca_shiny/
3. Survival Analysis: https://crsbanerji.shinyapps.io/survival_sim/

## Acknowledgements

CSRB was supported by the Turing-Roche Partnership. MVC was supported by the EPSRC Centre for Doctoral Training in Sustainable Chemical Technologies (EP/L016354/1) and Friends of FSH research (Project: An in-silico approach to understanding DUX4 expression). JP was supported by Muscular Dystrophy UK (19GRO-PG12-0493) and is currently by the FSHD Society (FSHD-Winter2021-4491649104). MG was supported by the Medical Research Council (MR/S002472/1) and now by SOLVE FSHD. The Zammit lab was also generously supported by Association Française contre les Myopathies.

## Additional information

### Funding

| Funder | Grant reference number | Author |
| --- | --- | --- |
| EPSRC Centre for Doctoral Training in Sustainable Chemical Technologies | EP/L016354/1 | Matthew V Cowley |
| Friends of FSH Research | | Matthew V Cowley |
| Muscular Dystrophy UK | 19GRO-PG12-0493 | Johanna Pruller |
| FSHD Society | FSHD-Winter2021-4491649104 | Johanna Pruller |
| Medical Research Council | MR/S002472/1 | Massimo Ganassi |
| Association Francaise contre les Myopathies | | Peter S Zammit |
| SOLVE FSHD | | Massimo Ganassi |
| The Turing-Roche Partnership | | Christopher RS Banerji |

| Funder | Grant reference number | Author |
|---|---|---|

The funders had no role in study design, data collection and interpretation, or the decision to submit the work for publication.

## Author contributions
Matthew V Cowley, Formal analysis, Investigation, Methodology, Writing – review and editing; Johanna Pruller, Conceptualization, Formal analysis, Investigation, Methodology, Writing – review and editing; Massimo Ganassi, Conceptualization, Writing – review and editing; Peter S Zammit, Conceptualization, Supervision, Funding acquisition, Methodology, Writing – review and editing; Christopher RS Banerji, Conceptualization, Data curation, Software, Formal analysis, Supervision, Funding acquisition, Investigation, Methodology, Writing – original draft, Project administration, Writing – review and editing

## Author ORCIDs
Matthew V Cowley ⓘ http://orcid.org/0000-0002-5258-8024
Massimo Ganassi ⓘ http://orcid.org/0000-0003-3163-9707
Peter S Zammit ⓘ http://orcid.org/0000-0001-9562-3072
Christopher RS Banerji ⓘ http://orcid.org/0000-0002-4373-7657

## Decision letter and Author response
Decision letter https://doi.org/10.7554/eLife.88345.sa1
Author response https://doi.org/10.7554/eLife.88345.sa2

# Additional files

## Supplementary files
• Supplementary file 1. Demographics of Facioscapulohumeral muscular dystrophy (FSHD) patients assayed in scRNAseq data. For each FSHD patient described in the scRNAseq dataset of *van den Heuvel et al., 2019*, we list sex, diagnosis, genotype, number of DUX4 +ve/-ve cells, and % of DUX4 +ve cells. For patients FSHD1.1 and FSHD2.1 where sufficient data was available, patient-specific estimates of $V_D$ are also given.

• Supplementary file 2. Promoter switching model parameter estimates for DUX4 target genes in patients FSHD1.1 and FSHD2.1. For each of the eight DUX4 target genes considered the three parameters underlying the promoter switching model ($k_a^n/\delta^n$, $k_i^n/\delta^n$, and $v_0^n/\delta^n$) derived from DUX4 +ve and DUX4-ve cells separately for patients FSHD1.1 and FSHD2.1 separately are presented.

• MDAR checklist

## Data availability
All data generated or analysed during this study are publicly available or included in the manuscript, all code employed is published as part of our shiny app at 3 public domain URLs listed in the manuscript, and available at GitHub: https://github.com/MVCowley/in-silico-FSHD-muscle-fiber-tools, copy archived at *Cowley, 2023*.

The following previously published datasets were used:

| Author(s) | Year | Dataset title | Dataset URL | Database and Identifier |
|---|---|---|---|---|
| van den Heuvel A, Mahfouz A, Kloet SL, Balog J, van Engelen BGM, Tawil R, Tapscott SJ, van der Maarel SM | 2018 | Single-cell RNA sequencing in patient-derived primary myocytes for facioscapuhumeral muscular dystrophy | https://www.ncbi.nlm.nih.gov/geo/query/acc.cgi?acc=GSE122873 | NCBI Gene Expression Omnibus, GSE122873 |
| Jiang S, Williams K, Kong X, Zeng W, Nguyen NV, Ma X, Tawil R, Yokomori K, Mortazavi A | 2020 | Single-nucleus RNA-seq identifies divergent populations of FSHD2 myotube nuclei | https://www.ncbi.nlm.nih.gov/geo/query/acc.cgi?acc=GSE143492 | NCBI Gene Expression Omnibus, GSE143492 |

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
