## [Editor Report]

To provide a logical answer to the over-expressed DUX4 in FSHD, the authors took a sophisticated mathematical modeling approach and applied it to empirical data. The approach successfully predicts behaviors of proteins and cells, thereby suggests a model for pathogenicity. The result poses a potential to be expanded to understand molecular dynamics of other mutation-mediated rare diseases.

---

## [Decision Letter]

[Editors' note: this paper was reviewed by Review Commons.]

---

## [Author Response]

Reviewer #1 (Evidence, reproducibility and clarity (Required)):Banerji and colleagues measure DUX4 and target gene expression over a time course in doxycycline-inducible myoblasts to estimate kinetic parameters and rates underlying the transition of non-expressing cells through DUX4-expressing cells to cell death. They then use these parameters to model the rate of appearance of DUX4+ cells, DUX4 target gene expression, etc. in cells from FSHD patients, and derive a model that predicts that over 100 days around one fourth of cells die while less than 1% of cells express DUX4 or its target genes at any given time. This is somewhat similar to what is seen in FSHD patients, where DUX4 expression is infrequent in cultured cells, while patients eventually have substantial muscle loss. The experiments are well-designed and explained clearly.

We thank the reviewer for their kind comments on our manuscript and for acknowledging the similarity between our model simulation and both cell biological and clinical features of FSHD.

Reviewer #1 (Significance (Required)):The primary significance of this study is that the field has a sense that the damage seen in patient muscle is not congruent with the low expression of DUX4 in patients, and the model showing many cells dying with only a few cells expressing DUX4 at any given time suggests that overall damage can be greater than that observed in any particular snapshot.However, it is important not to conflate low frequencies of DUX4+ nuclei in cultured myoblasts with "DUX4 being difficult to detect" as in p 12, Discussion. DUX4 is difficult to detect, indeed basically not detected, in muscle biopsy specimens, but in cells in vitro, DUX4 is fairly easy to detect, albeit in quite low numbers of cells. Since the study evaluates cells in vitro, it is important to make clear that the situation in vivo is qualitatively different from that seen in vitro, namely DUX4 not being detected, and the authors should clarify this importance difference.

We appreciate this important point that DUX4 detection is extremely challenging in FSHD patient muscle biopsies, compared to in vitro cell culture of FSHD myoblasts and myotubes, a topic we considered at length in our recent review (Banerji and Zammit 2021, https://doi.org/10.15252/emmm.202013695).

Detection of DUX4 mRNA in muscle biopsies is highly variable requiring nested RT-qPCR, and protein detection is even less robust, though this has been achieved via western blot in less affected FSHD muscle (Tassin et al., 2013, DOI: 10.1111/j.1582-4934.2012.01647.x) and using a proximity ligation assay (Beermann et al. 2022, doi: 10.1186/s13104-022-06054-8). DUX4 target gene expression is detected in inflamed FSHD muscle but less robustly in non-inflamed muscle, indicating the recent presence of DUX4 protein, and also implying that a systemic component may activate DUX4 expression in vivo.

Conversely in cell culture of FSHD muscle cells, DUX4 mRNA is typically found in 0.5-3.8% of myonuclei (van den Heuvel et al., 2019, https://doi.org/10.1093/hmg/ddy400) and protein in around 1/1000 myonuclei (Snider et al., 2010, https://doi.org/10.1371/journal.pgen.1001181). Arguably these levels are still very low and DUX4 is still ‘difficult to detect’ in vitro, for example DUX4 expression is never seen above the limit of detection in bulk RNA-seq of FSHD patient cultured myoblasts. However, we fully appreciate that DUX4 is ‘easier to detect’ in vitro versus in vivo.

Regardless, anti-DUX4 therapy is currently the most heavily funded approach to FSHD treatment, and all anti-DUX4 therapies currently under consideration were first investigated in the myoblast in vitro setting. Thus, the aim of our model is to provide an additional level of in silico screening for anti-DUX4 therapy, before progressing to in vitro investigation.

Full consideration of DUX4 expression in vivo (while an exciting prospect), would require significantly more data than is currently available (relating to inflammation, vascularity and other systemic factors) and a highly complex model which is beyond the remit of this investigation.

We will clarify in the manuscript that our model applies in vitro, highlighting differences with in vivo and emphasising that the model is limited for understanding DUX4 expression in FSHD muscle biopsies.

**Changes to manuscript:**

Page 13, Line 443: We have removed the statement: ‘with DUX4 mRNA, protein and target gene accumulation all difficult to detect’.

Page 13, Lines 473-481: We have added the following to the Discussion:

‘We also limited our model to differentiated muscle culture, when DUX4 expression is more robust. DUX4 mRNA is typically detected in a small proportion (0.5-3.8%) of FSHD myoblasts in vitro^16,17,21^, and protein expression is similarly rare (0.1%)^18^. The situation is qualitatively different in FSHD muscle biopsies, where DUX4 mRNA detection is highly variable and DUX4 protein detection is extremely rare. DUX4 target gene expression is detected in inflamed FSHD muscle biopsies but less robustly in non-inflamed muscle, implying that a systemic component may also activate DUX4 expression in vivo^2,48,49^. Despite the added complexity of DUX4 expression in the muscle biopsy setting, all anti-DUX4 therapies currently under consideration were first investigated in myogenic cells in vitro.’

A second reason for caution in extrapolating correlates from the in vitro model to the disease process in muscle tissue is that in vivo there is a continual source of replacement cells, as the authors have shown in a previous study. Have the authors attempted to model a situation in which new cells are provided into the system at some rate, related to the amount of death occurring at different times? Although the authors mention that the static cell number is a limitation of the model, it would be valuable to revisit or explore this idea in the Discussion section, if only to provide the reader with a more pragmatic perspective.

We are focused specifically on modelling the in vitro differentiated myogenic cell setting to provide an in silico pre-screening tool for assessing anti-DUX4 therapy, rather than attempting to model the full pathology in vivo, which given data limitations, is beyond the scope of our study. As the reviewer notes, we highlight this limitation relating to proliferation when we introduce the model in the results stating: ‘Cells are assumed not to proliferate over the evolution of the model. We restrict applications to differentiating cells which have exited the cell cycle.’

As the reviewer likely appreciates, DUX4 expression in proliferating cells differs significantly from differentiated cells. The single nuclear and single cell RNA-seq data we employ to derive the parameters underlying DUX4 and DUX4 target transcription rates are from non-proliferating myocytes differentiated for 3 days. Hence it is not appropriate to include proliferation in this model, as the transcription rates we use will not be accurate for proliferating cells.

Introducing proliferation will also require us to estimate the proliferation rate of FSHD myoblasts, and how it is impacted by DUX4 expression, as well as how DUX4/target transcription rates change as the proliferating cells differentiate. Though this is an interesting application there is not sufficient data available to produce this wider scale model at present.

We thus restrict application of our model to the non-proliferating differentiated setting, which can be easily accessed in vitro to screen anti-DUX4 therapy.

We appreciate the observation that in vivo, there will be addition of new cells during muscle regeneration, and will include this in discussion in the updated manuscript. We will also expand on our discussion of what data would be required to include proliferation in our model in the revised manuscript.

Changes to manuscript:

Page 14, Lines 500-503: We have added the following to the Discussion:

‘Moreover, new data describing scRNAseq of proliferating FSHD primary myoblasts, alongside quantification of how DUX4 and DUX4 target gene expression impacts myoblast proliferation rates, would allow natural extension of our model to understand DUX4 expression during FSHD myoblast proliferation.’

The second part of the paper models presence of DUX4 in nuclei based on diffusion from expressing to non-expressing nuclei, and characterizes this as the activity of "an infectious agent, able to spread from one nucleus to another by adapting epidemiological compartment models". The relationship to an infection process is probably not the ideal way to characterize this process, because an infection implies the setting up of new sites of production of the agent, DUX4, where what is really happening is that DUX4 diffusing into these other nuclei isn't leading to more DUX4 production, it is just diffusing into nearby nuclei and accumulating there. Unless I am misunderstanding, the authors are simply showing that a larger number of nuclei will be positive in a system in which cells are fusion products having many nuclei than in a system in which all nuclei are isolated within their own cells.

The term ‘infectious agent’ is only an analogy and implies some preconceptions. The reviewer interprets our infectious agent model as: ‘an infection implies the setting up of new sites of production of the agent, DUX4, where what is really happening is that DUX4 diffusing into these other nuclei isn't leading to more DUX4 production, it is just diffusing into nearby nuclei and accumulating there. Our bespoke model is not based on a direct mapping of e.g., viral infection to our setting. In our case, the ‘infectious agent’ DUX4 causes harm but does not replicate. DUX4 protein is a transcription factor and thus activates the expression of target genes (not including DUX4). In our model, DUX4 can be seen as the ‘infectious agent’, and expression of DUX4 target genes can be seen as the ‘infection’, which leads to cell death. DUX4 can either stay in the ‘infected’ cell until it dies or diffuse to another cell to spread the ‘infection’.

The reviewer interprets the results of our model as ‘simply showing that a larger number of nuclei will be [DUX4] positive in a system in which cells are fusion products having many nuclei than in a system in which all nuclei are isolated within their own cells’.

In fact, we make several observations: firstly we compare single cell RNA-seq of unfused myocytes to single nuclear RNA-seq of fused muti-nucleated myotubes and find that the latter has a greater proportion of cells expressing DUX4 target genes (i.e. more infection). The proportion of DUX4 positive cells (i.e. the amount of infectious agent produced) is similar in the two settings. We posit that this difference in DUX4 target gene positive cells (infection) may be due to DUX4 protein diffusing between cells in the myotube syncytium and activating the targets in neighbouring nuclei (i.e., greater mobility of the infectious agent/loss of quarantine). Note that there are many other possibilities, such as fusion causing an increase in DUX4 transcription/mRNA stability etc.

We find that by introducing a DUX4 protein diffusion term (loss of quarantine) into our model we can completely explain the difference between DUX4 target gene expression (infection) in real data of unfused versus fused myonuclei, despite identical levels of DUX4 (infectious agent). This is a novel finding providing evidence for DUX4 protein diffusion in syncytial myonuclei, which represents an often overlooked therapeutic target/consideration in FSHD. We also provide the first explicit quantification of this diffusion rate via our model as well as a framework for investigators to understand how modifying this parameter can impact DUX4 induced myotoxicity.

We will better define our ‘infectious agent’ analogy and clarify our interpretation in the updated manuscript.

Changes to Manuscript:

Page 9, Lines 338-343: We add the following clarification when we introduce our infection analogy model in the Results:

The term ‘infect’ is only an analogy and our model is not based on a direct mapping of e.g., viral infection to our setting. In our case, the ‘infectious agent’ DUX4 causes harm but does not replicate. DUX4 protein is a transcription factor and thus activates the expression of target genes. In our model, DUX4 can be seen as an ‘infectious agent’, and expression of DUX4 target genes can be seen as the ‘infection’, which leads to cell death. DUX4 can either stay in the ‘infected’ cell until it dies or diffuse to another cell to spread the ‘infection’.’

Page 14, Lines 514-525: We have edited our interpretation of the ‘infectious agent’ model results in the Discussion:

‘DUX4 has also been proposed to spread through the myofibre syncytium from its originator nucleus to ‘infect’ DUX4 naïve nuclei, bypassing the need for a rare expression burst and accelerating pathology^26,48^. The proportion of DUX4 target positive cells is greater in snRNAseq of syncytial FSHD myotubes than scRNAseq of unfused myocytes, while the proportion of DUX4 positive cells is comparable, supporting a syncytial diffusion mechanism. We model

DUX4 in FSHD myotubes as an infectious agent, able to spread from one nucleus to another (but not replicate) by adapting epidemiological compartment models, which we package as a cellular automaton to provide a realistic myotube surface on which to monitor DUX4 expression. Incorporation of a diffusion term is sufficient to account for the difference in DUX4 target gene +ve/-ve nuclear proportions, between syncytial FSHD myotubes and unfused FSHD myocytes, while maintaining comparable proportions of DUX4 +ve cells. We thus demonstrate that the theoretical hypothesis of DUX4 syncytial diffusion is compatible with biological data, and provide an estimate for the DUX4 diffusion rate.’

Reviewer #2 (Evidence, reproducibility and clarity (Required)):An in silico FSHD muscle fibre for modelling DUX4 dynamicsSummaryThis paper studies and mathematically models the stochastic process of presence or absence of DUX4 expression in individual myocytes in FSHD, which underlies the very small proportion of individual cells which do express DUX4. The paper is valuable in providing a mathematical basis for modelling this, and hence a basis by which to study the effects of other factors or potential therapeutic interventions which may influence the overall proportion of cells which do express.

We thank the reviewer for the kind comments on our manuscript and recognition of its value to design/screening of potential therapeutics for FSHD.

To understand the paper fully, requires a familiarity with mathematical and statistical reasoning which will be beyond many potential readers, including this reviewer, but I hope that the following specific comments may still be helpful.

We appreciate the reviewer’s candour regarding their understanding of the more advanced mathematical aspects the paper. Interdisciplinary work such as ours may introduce concepts which are unfamiliar to some readers.

As the reviewer will appreciate, understanding a complex process like DUX4 expression requires new and more sophisticated approaches, having largely eluded the conventional approaches to date. The main mathematical tools employed in our work derive from the wellestablished stochastic gene expression and epidemiological compartment models. We have endeavoured to make the mathematics self-contained to facilitate understanding and will edit to further improve accessibility.

Changes to Manuscript:

Multiple edits have been made in response to the helpful comments of both reviewers which has improved accessibility of the more technical aspects of the manuscript.

Major Comments1. A potential major concern is that the modelling seems to be based largely on combined data from 27 DUX-4 positive myocytes out of 5133 myocytes from 4 FSHD patients described by Van Den Heuvel et al., 2019 (ref 17), but with 2 being FSHD1 and 2 FSHD2, and without the 2 patients within each FSHD type being matched for D4Z4 fragment length (RU = 'residual units'), which can be expected to be a major influencing factor on the threshold for expression in any one myocyte upon which the stochastic process acts. Thus, one can expect lower RU number to show a greater proportion of expressing cells within FSHD1, and to some extent similarly within FSHD2 (although in FSHD2, different SMCHD1 mutations will also have different 'strengths'). The 4 patients were : 2x FSHD1 (3RU ; 6RU); 2x FSHD2 (12RU ; 18 RU). In the paper of Den Heuvel et al. it is evident that the DUX4-positive cell count differs between these 4 cell lines very much as might be anticipated from their D4Z4 RU number. Thus (Figure 2 in that paper) patient 1.1 (3RU) has 12 positive cells, whereas patient 1.2 (6RU) has only 2 positive cells. Patient 2.1 (SMCHD1 exon 37 mutation, and 12 RU) has 11 positive cells, whereas Patient 2.2 (SMCHD1 exon 10 mutation, and 18 RU) has only 2 positive cells.Therefore, if the validity and accuracy of the modelling here would be affected by genetic heterogeneity, the authors should present their calculations and analyses based only on the 2 patients who account for 23/27 of the positive cells, and furthermore, that the calculations and modelling are performed and presented for those 2 patients separately.

The reviewer raises the important point that 2/5 of the parameters of our model were derived from the single cell RNA-seq data set of van den Heuvel et al., which comprises 4 FSHD patients of various genotype. The reviewer remarks that the number of DUX4 positive cells reported for each patient in the van den Heuvel data, differs according to genotype, in line with the known inverse association between DUX4 expression and D4Z4 short allele length. Figure 2E in van den Heuvel et al., to which the reviewer refers, does not present absolute DUX4+ cell counts, but rather ‘DUX4 affected’ cell counts, based on the expression of 67 DUX4 target genes. The numbers in that figure thus differ from ours. We caution that the number of ‘DUX4 affected’ cell counts must be normalised by the total number of cells assayed per patient, to obtain a ‘DUX4 affected’ proportion for each patient. However, we appreciate the reviewers point.

A concern is raised that we have pooled all 4 FSHD patients to derive our parameter estimates, essentially averaging over genotypes. The two parameters we derived from this data are ‘the average DUX4 transcription rate, and ‘the average DUX4 target gene transcription rate.

The average DUX4 target gene transcription rate was derived from consideration of DUX4 target gene expression in the 27 FSHD cells already expressing DUX4 mRNA. Since the capacity of DUX4 to activate its target genes is not known to depend on genotype, we feel it is appropriate to pool the patients to estimate this parameter.

For the average DUX4 transcription rate however, we do appreciate the reviewer’s concern that patient genotype may impact the estimate. The reason for pooling all patients is partly statistical. As there are so few cells expressing DUX4 in each patient, the gradient descent algorithm we employed to estimate the 3 rate parameters for the promoter switching model of DUX4, may fail to converge for every patient separately. By pooling patients, we obtained an estimate that may not be true for an individual but represented the ‘average’ transcription rate of DUX4 in these 4 FSHD patients, hence our careful use of wording when defining the average DUX4 transcription rate parameter.

Our in silico model of DUX4 expression is proposed as a pre- in vitro screening tool to guide anti-DUX4 therapy for the general population, not a digital twin of a single FSHD patient, and so an average DUX4 transcription rate derived from pooling available data is optimal.

This average measure is not irrelevant in light of genetic variation and in fact may be more informative for anti-DUX4 therapeutics intended for a general population, but less informative for genotype stratified therapeutic design. Genotypic stratification by D4Z4 repeat length was not performed in recent FSHD clinical trials for anti-DUX4 therapy (e.g., losmapimod: NCT04003974), which recruited patients with all pathogenic D4Z4 repeat lengths, motivating an average over genotype design in the first instance.

We will elaborate in detail on the reasons for patient pooling throughout the manuscript.

We will attempt the analysis the reviewer suggests i.e. ‘the authors should present their calculations and analyses based only on the 2 patients who account for 23/27 of the positive cells…separately’. However, this is conditional on algorithm convergence given the smaller number of cells, and we must caveat that in doing so we will have eliminated 2 patients purely on the basis of low DUX4 expression and so will over-estimate the average DUX4 transcription rate for FSHD patients in general.

New analysis:

We are pleased to report that our gradient descent algorithm converged enabling estimation of the DUX4 transcription rate V_D_ for the 2 patients with the largest number of DUX4+ve cells, as the reviewer requested. Our pooled estimate of V_D_ based on all 4 patients was 0.0021/hour. The estimate from patient FSHD1.1 described by van den Heuvel et al., was 0.00374/hour and from patient FSHD2.1 was 0.00096/hour. We note that the average value for these two patients is 0.00235/hour, comparable but slightly higher than our pooled estimate across all 4 patients. This is in line with our expectation that eliminating 2/4 patients on the basis of low DUX4 expression may contribute to an over-estimate of the true average DUX4 transcription rate.

Personalised estimates of the DUX4 target gene transcription rate V_T_ could not be computed as this required us to estimate parameters of the promoter switching model for the 8 DUX4 target genes, from consideration of only 19 DUX4 +ve cells for patient FSHD1.1 and 5 DUX4+ve cells for patient FSHD2.1. For 3/8 genes for patient FSHD1.1 and 1/8 genes for patient FSHD2.1 our gradient descent algorithm failed to converge to MLEs of the parameters, preventing calculation of V_T_. We explore the parameters which did converge in response to point 9 below.

Given this and the fact that we are constructing a general model rather than a personalised model (for which there is insufficient data), we believe it is most appropriate to present all subsequent calculations in the manuscript using the pooled estimates of both V_D_ and V_T_.

However, for the editor’s interest, we also ran our compartment model employing the values of V_D_ separately for patients FSHD1.1 and FSHD2.1, with the pooled value V_T_. In Author response image 1 and 2 we have recreated figure 4A for each patient, comparing model simulated cell proportions to the true cell proportions for each patient in the scRNAseq data. We see that for each patient the simulated proportions are statistically indistinguishable from the data as assessed by Chi Squared test, as for the pooled model:

**Author response image 1. sa2fig1:** Model simulation with VD derived for patient FSHD1. 1, using appropriate starting cell numbers for this patient: Model vs Data Chi Squared p=0.99.

**Author response image 2. sa2fig2:** Model simulation with VD derived for patient FSHD2. 1, using appropriate starting cell numbers for this patient: Model vs Data Chi Squared p=0.99.

Though encouraging, these models are not fully personalised as they use a pooled V_T_, to avoid the risk that readers interpret this model as fully personalised, we choose not to include these data in the published manuscript and focus instead on the general model derived from only pooled estimates.

Changes to Manuscript:

Page 6, Lines 196-206: We have updated the Results to include:

‘As the total number of DUX4 positive cells is low, we pooled data from the 4 FSHD patients to allow robust estimation of the average DUX4 transcription rate for this patient cohort. We note, however, that distinct FSHD genotypes likely underlie different DUX4 transcription rates. The majority of DUX4 +ve cells (23/27) were found in two FSHD patients for this scRNAseq data. For patient FSHD1.1 19/2226 (0.85%) cells were DUX4 +ve and for patient FSHD2.1 5/1283 (0.39%) cells were DUX4 +ve (Supplementary File 1). To investigate variability in V_D_, we derived individual estimates for these two ‘higher’ DUX4 expressing patients, for FSHD1.1 V_D_ = 0.00373/hour, while for FSHD2.1 V_D_ = 0.000960/hour. The pooled estimate of V_D_ is thus comparable in order of magnitude with that of individual patients. To fully utilise the available data and prevent limiting our model to only ‘higher’ DUX4 expressing patients, we employ the pooled estimate of V_D_ = 0.00211/hour for the remainder of our calculations.’

Page 14, Lines 486-499: We have updated the Discussion to include:

‘FSHD is a rare disease and public data sets describing scRNAseq and snRNAseq of patient derived primary myocytes are currently limited to those used in this study. Consequently, we have pooled data describing 4 individual patients of different FSHD genotype to estimate two of the parameters of our model: V_T_ and V_D_. DUX4 expression levels depend on D4Z4 repeat length and methylation status, and can differ between cell lines isolated from different FSHD patients^14,50^, as well as between genetically identical cell lines isolated from the same mosaic FSHD patient^51^. We computed patient specific estimates of V_D_ for 2/4 patients with sufficient data and found these comparable to the pooled estimate. By pooling patients, we obtained parameter estimates that may not be true for an individual but represented the ‘average’ parameter values across these 4 FSHD patients. Our in silico model of DUX4 expression is proposed as a pre- in vitro screening tool to guide anti-DUX4 therapy for the general population. However, as more data on FSHD is generated our models will evolve. In particular, as higher volumes of scRNAseq data describing FSHD patients with different genotypes become available, the model can be updated to facilitate genotype stratified and even personalised antiDUX4 therapy design.’

In order to be of value for assessing potential interventions more widely in FSHD, the authors would then need, in further publications, to extend their analysis similarly to genetically homogeneous myocyte sets with other RU number.

We agree with the reviewer that for genotypically stratified therapeutic design, based on D4Z4 short allele length, personalised or genotype specific estimates of the DUX4 transcription rate should be calculated. However, as the reviewer clearly appreciates, developing such FSHD digital twins, though exciting, would require a much larger library of single cell RNA-seq datasets describing FSHD patients of various genotypes than is currently available.

We will explore this in an updated discussion.

Changes to Manuscript:

See relevant changes to discussion in response to above point.

Specific points.2. Introduction 4th paragraph: – 'Single cell and single nuclear transcriptomic studies find only 0.5-3.8% of in vitro differentiated FSHD myonuclei express DUX4 transcript16,17. Immunolabelling studies only detect DUX4 protein in between 0.1-5% of FSHD myonuclei18,19.' The authors should comment whether there is any evidence from the data in these references, or from other publications, regarding differences in these proportions by FSHD diagnosis (I or II) or by D4Z4 residual number?.

We will expand discussion in the updated manuscript of the result (explained by the reviewer) that the number of DUX4 affected cells found in the single cell RNA-seq dataset of van den Heuvel may correlate with genotype. We will contrast this with the observation that immortalised isogenic FSHD myoblast clones isolated from a single FSHD patient can also have a very wide range of DUX4 expression (e.g., Krom et al., doi: 10.1016/j.ajpath.2012.07.007). This will highlight the mix of poorly understood genetic and epigenetic factors impacting DUX4 expression and further motivate an average model in the first instance.

Changes to Manuscript:

See relevant changes to discussion in response to the first point raised by the reviewer, in particular:

‘DUX4 expression levels depend on D4Z4 repeat length and methylation status, and can differ between cell lines isolated from different FSHD patients^14,50^, as well as between genetically identical cell lines isolated from the same mosaic FSHD patient^51^.’

3. Introduction 5^th^ paragraph: – Importantly, a recent phase 2b clinical trial of the DUX4 inhibitor losmapimod failed to reach its primary endpoint of reduced DUX4 target gene expression in patient muscle, despite improvement in functional outcomes23. Please state here whether or not there was any evidence that DUX4 expression itself was reduced in this trial. It would help if the authors could also add here an interpretive comment as to whether this indicates that the current presumed target genes for DUX4 are not in fact the ones important for FSHD, or whether their expression might be retained from compensatory upregulation by other upstream regulators. The authors should expand briefly also in the introduction on the evidence for toxicity of target gene expression.

We are very happy to include a discussion of these points, which we explore in considerable detail in our recent review of DUX4 in FSHD (Banerji and Zammit 2021, https://doi.org/10.15252/emmm.202013695). Briefly, the losmapimod trial did not publish data showing they measured DUX4, as Reviewer #1 pointed out DUX4 is highly difficult to detect in patient muscle biopsies and dynamic change following anti-DUX4 therapy has certainly never been observed. Expression of four DUX4 target genes was intended as a surrogate measure of DUX4 activity, reasons for its lack of change in this trial is very much a topic of open debate, which we will outline in an updated manuscript.

Changes to Manuscript:

Page 2, Lines 64-70: We include the following statement in the Introduction:

‘Given the challenge of detecting DUX4 in muscle biopsies^4^ it is unsurprising that no data was published relating to DUX4 expression changes during the losmapimod clinical trial. An understanding as to why losmapimod did not alter expression of DUX4 targets in patient muscle biopsies is also lacking, but hypotheses include heterogeneity in muscle sampling, low baseline levels of DUX4 targets and thus limited dynamic range, slow reversibility of DUX4 induced epigenetic changes on target gene promoters and losmapimod having limited impact on DUX4 transcriptional activity in vivo.’

4. Results 2nd paragraph: – We first describe the compartment model. FSHD single myocytes can express DUX4 and therefore DUX4 target genes; DUX4 target gene expression leads to cell death19,27,28. Please indicate whether the evidence in these studies is independent of DUX4. Ie. Is there independent evidence that it is definitely the target gene expression, rather than DUX4 expression per se, that leads to cell death, especially if losmapimod improves function by inhibiting DUX4, but has no impact on these target genes.

This is an important point raised by the reviewer and one that we have rigorously investigated (Knopp et al., 2016, https://doi.org/10.1242/jcs.180372).

The c-terminus of DUX4 is a potent transcriptional activator and is required for DUX4 target gene activation, while an inverted centromeric D4Z4 repeat unit encodes a version of DUX4 lacking the c-terminus, named DUX4c. We constructed a library of DUX4 expression constructs encoding: DUX4, DUX4c, tMALDUX4 (DUX4 with the c-terminus removed), DUX4-VP16 (DUX4 with the c-terminus replaced by the VP16 transactivation domain) and DUX4-ERD (DUX4 with the c-terminus replaced by the engrailed repressor domain). In a series of experiments outlined in Knopp et al., we clearly demonstrated that only DUX4 and DUX4-VP16 could activate DUX4 targets in C2C12 murine myoblasts and that only these constructs could induce apoptosis. Our prior work provides this clear link between DUX4 target gene expression and cell death. We will include discussion of this point in the updated manuscript.

Changes to Manuscript:

Page 4, Lines 109-110:We include the following statement in the Results:

‘We have previously demonstrated, via a library of DUX4 expression constructs, that DUX4 target gene activation is also necessary for DUX4-induced cell death^31^.’

5. Results 2nd paragraph Item 4 : – R(t) – a resigned state where the cell expresses no DUX4 mRNA but does express DUX4 target mRNA (DUX4 -ve/Target gene +ve: i.e. a historically DUX4 mRNA expressing cell) Since the mRNA is from the target genes, why should it be produced only in response to DUX4 expression ? ie. Please indicate the evidence to say that these must be 'historically DUX4 mRNA expressing cells', rather than target gene expression in response to other factors, or even autonomously.

The reviewer raises the point that the 8 DUX4 target genes we investigate may be expressed in a DUX4 independent manner, making DUX4 target gene +ve, DUX4 -ve cells, potentially not historic DUX4 expressing cells. We selected these 8 genes as they are confirmed DUX4 targets by ChIP-seq and have been identified as upregulated by DUX4 in every transcriptomic analysis of DUX4 over-expressing human myoblasts (Banerji and Zammit 2021, https://doi.org/10.15252/emmm.202013695). We provide further evidence for these genes indicating historic DUX4 in these cells: first, we never find a single transcript for any of these 8 target genes in 1914 single myocytes and 77 single nuclei from control individuals, suggesting that their expression requires an FSHD genotype (and thus likely DUX4 expression). Second, our investigation of these genes in the single cell RNA-seq data set of van den Heuvel demonstrated that in the presence of DUX4, expression of all 8 genes significantly increase (Figure 3D), indicating activation by DUX4.

While this is evidence that the 8 targets are specific to DUX4, we accept that it is not 100% conclusive and these genes may also be induced by some other factor related to the FSHD genotype. We will thus highlight this possibility of other modes of expression in the discussion.

Changes to Manuscript:

Page 13, Lines 458-472: We include the following statement in the Discussion:

‘We further assumed that cells expressing our 8 DUX4 target genes are historic DUX4 expressing cells. We selected these 8 genes as they are confirmed DUX4 targets by ChIP-seq and have been identified as upregulated in every transcriptomic analysis of DUX4 overexpressing human myoblasts^4^. Tissue expression patterns of these 8 genes are not well characterised outside of the FSHD context, however their expression has been reported during zygotic genome activation^46^ and in testicular tissue^1,2^, both settings where DUX4 is physiologically expressed. We found further evidence for expression of these genes indicating historic DUX4 in this study. First, we never find a single transcript for any of these 8 target genes in 1914 single myocytes and 77 single nuclei from control individuals, suggesting that their expression requires an FSHD genotype (and thus likely DUX4 expression). Second, our investigation of these genes in scRNAseq data of FSHD myocytes demonstrated that in the presence of DUX4, expression of all 8 genes significantly increases, indicating activation by DUX4. While this is evidence that the 8 targets are specific to DUX4, it is possible that they may also be induced at a much lower level, by some other factor related to the FSHD genotype and thus a small proportion of DUX4 target gene +ve cells in this study may not be historically DUX4 +ve.’

6. Results – Estimating the kinetics of DUX4 mRNA – 2nd paragraph: – DUX4 expression was induced with 250 ng/ml doxycycline for 7 hours.…. Might some of the laboratory detail here be better placed in the 'Methods' section ?

We will move some methodological detail to the methods as suggested.

Changes to Manuscript:

Details on concentrations of doxycycline and duration of induction have been moved from Results page 5 to Materials and methods page 16.

7. Results – Estimating the kinetics of DUX4 mRNA – 4th paragraph: – '…from 4 FSHD patients (2 FSHD1 and 2 FSHD2).' Please give additional information about these patients – ie. Age, sex, severity (or age-onset), and genetic results – perhaps best done as a Table in this paper rather than only by reference back to the Van den Heuvel et al. paper (see main comment 1 above).

We will provide this information in a table as the reviewer suggests.

Changes to Manuscript:

We now include a supplementary table: Supplementary File 1 detailing for each FSHD patient described by van den Heuvel et al., age, sex, genetic results as well as DUX4 +ve/-ve cell counts and where appropriate patient specific estimates of V_D_. We note that details of patient severity were not recorded by van den Heuvel et al.

Page 6, Lines 188-189: This table is referenced in the Results:

‘Patient demographics, genotypes and DUX4+/-ve cell counts are displayed in Supplementary File 1.’

A legend for this table is included on page 24 lines 870-873:

‘Supplementary File 1: Demographics of FSHD patients assayed in scRNAseq data.

For each FSHD patient described in the scRNAseq data set of van den Heuvel et al., we list sex, diagnosis, genotype, number of DUX4 +ve/-ve cells and % of DUX4+ve cells. For patients FSHD1.1 and FSHD2.1 where sufficient data was available, patient specific estimates of V_D_ are also given.’

8. Results – Estimating kinetics of DUX4 target activation – 2nd paragraph: – '8 DUX4 target genes……expression was restricted to FSHD cells/nuclei and never observed in controls.' So are these genes normally only expressed in the zygote , rather than post-birth (except perhaps the testis)? The authors should comment on this.

The 8 DUX4 target genes are not well characterised in terms of their expression pattern in other tissues. We are confident that in our data sets they are not expressed in healthy myocytes and myotubes, but it is difficult to expand confidently on other tissues due to lack of data. We will include comment on what is known about their expression pattern during zygotic genome activation and in testicular tissue, but this cannot be an exhaustive description of their expression. For example, DUX4 has been detected in keratinocytes, osteoblasts and lymphocytes, but the target genes have not been characterised in such settings. We still have much to learn about the expression of these genes outside of the muscle setting.

Changes to Manuscript:

See change in Discussion on page 13, outlined in response to point 5 above, in particular the statement:

‘Tissue expression patterns of these 8 genes are not well characterised outside of the FSHD context, however their expression has been reported during zygotic genome activation^46^ and in testicular tissue^1,2^, both settings where DUX4 is also expressed.’

9. Results – Estimating kinetics of DUX4 target activation – 4nd paragraph: – In the presence of DUX4 mRNA the proportion of time the promoters of the 8 DUX4 target genes remained in the active state,….significantly increased. It would be interesting to know if any of these temporal dynamic descriptors of DUX4 target promoters differ consistently between sample II-2 and II-1 and in the same direction between I-2 and I-1. ie. Might the same factors (eg. D4Z4 copy number) which may affect presence/absence of DUX4 also affect other cellular measures of DUX 4 ?

The reviewer raises an interesting point about genetic modifiers of DUX4 transcriptional

activation. As discussed in response to point 1, the DUX4 target gene transcription rate is derived from the 27 cells expressing DUX4. As there are so few cells expressing DUX4 in each patient (sometimes <3), the gradient descent algorithm we employ to estimate the 3 rate parameters for the promoter switching model of each DUX4 target, will fail to converge for every patient separately.

We will attempt analysis of the two patients which provide the majority of DUX4 positive cells separately, as the reviewer requested in point 1. If successful, we will investigate whether the derived DUX4 target gene expression parameters differ between these two patients as the reviewer suggests. While this limited data will not permit anything definitive about impact of D4Z4 repeat length on DUX4 target gene expression, it may highlight the existence of interindividual differences in such parameters.

New Analysis

We have attempted the analysis suggested by the reviewer: comparing the promoter switching model parameters for the 8 DUX4 target genes, between DUX4+ve and DUX4-ve cells separately for patients FSHD1.1 and FSHD2.1. Unfortunately, as remarked in our response to point 1 above, insufficient data prevented parameter estimation for 3/8 of the DUX4 target genes (*PRAMEF12*, *PRAMEF2* and *RFPL2*) for patient FSHD1.1 and 1/8 DUX4 target genes (*RFPL2*) for patient FSHD2.1. Consequently, our comparison between patients is limited to the genes for which we could derive parameter estimates.

Despite this limitation, for both patients FSHD1.1 and FSHD2.1 parameter differences observed between DUX4 +ve and DUX4-ve cells were consistent with those seen for all cells pooled.

In particular, as with the pooled estimates, the proportion of time the promoter of the target genes remained in the active state was higher in DUX4 +ve compared to DUX4 -ve cells for each patient separately. This achieved significance for patient FSHD2.1 (paired Wilcoxon *p=0.0156*, n=7 genes*)*, with borderline significance for patient FSHD1.1 (paired Wilcoxon *p=0.0625,* n=5 genes). The lower significance for FSHD1.1 may be related to the smaller number of genes for which we could derive parameter values.

Again, as with the pooled estimates, the transcription rate of the target genes again was higher in the DUX4-ve cells compared to DUX4+ve. This trend approached significance for FSHD2.1 (paired Wilcoxon *p=0.0781,* n=7 genes*)*, but was not significant for FSHD1.1 (paired Wilcoxon *p=0.35,* n=5 genes*)*.

Lastly as with the pooled estimates, the mean expression of the target genes was higher in DUX4+ve cells compared to DUX4-ve. This was significant for FSHD2.1 (paired Wilcoxon *p=0.0156,* n=7 genes*)* and approached significance for FSHD1.1 (paired Wilcoxon *p=0.0625,* n=5 genes*)*.

We include these data as supplementary information to the updated manuscript.

Changes to Manuscript:

Page 8, Lines 267-276: The following has been added to the Results:

‘As with our calculation of V_D_ data was pooled across 4 FSHD patients to calculate V_T_. We do not anticipate patient genotype to impact the average DUX4 target transcription rate, independently of its impact on the DUX4 transcription rate. However, to confirm our findings on the impact of DUX4 on target gene promoter dynamics, in a patient specific setting, we attempted calculation of the parameters underlying the promoter switching model for the 8 DUX4 target genes in DUX4+ve and DUX4-ve cells, for patients FSHD1.1 and FSHD2.1 separately. Due to the limited number of cells for each patient, personalised estimates for all 8 target genes could not be obtained. However, where patient specific estimates were obtained the direction of parameter differences in target genes, between DUX4 +ve and DUX4 -ve cells were in line with those of pooled estimates across 4 FSHD patients (Supplementary File S2).’

Page 24, Lines 874-878: A supplementary table describing the target gene parameter estimates is also provided to the manuscript alongside a legend:

‘Supplementary File 2: Promoter switching model parameter estimates for DUX4 target genes in patients FSHD1.1 and FSHD2.1.

For each of the 8 DUX4 target genes considered the three parameters underlying the promoter switching model (k_a_^n^/δ^n^,k_i_^n^/δ^n^ and v_0_^n^/δ^n^) derived from DUX4+ve and DUX4-ve cells separately for patients FSHD1.1 and FSHD2.1 separately are presented.’

10. Results – Compartment model simulation – 4th paragraph: – '…so that after 10 days 26.3% of cells had died.' The authors need to give data here also for controls. Ie. What proportion of cells die after 10 days in controls ?

As mentioned in the introduction of our model, we only model DUX4 dependent cell death. As control cells do not express DUX4 they cannot experience cell death by this mechanism. The reviewer can therefore interpret this result as excess death due to DUX4, i.e., the 26.3% of FSHD cells die which would not have died in control cells. We will clarify this point in an updated manuscript.

Changes to Manuscript:

Page 9, Lines 307-309: We have made the following change to the Results:

‘As expected, the number of cells in the DUX4 naive, susceptible state S(t) gradually decreased, while the number of dead cells due to DUX4 D(t) gradually rose, so that after 10 days 26.3% of cells had died as a consequence of DUX4 expression.’

11. Discussion – 1st paragraph: – ' However, DUX4 expression in FSHD muscle demonstrates a complex dynamic with DUX4 mRNA, protein and target gene accumulation all difficult to detect2,21. Understanding this complex dynamic is essential to the construction of optimal therapy' So, it presumably follows that comparison by known genetic or demographic modifiers of disease severity (such as D4Z4 copy number could be of fundamental importance in interpreting this dynamic and hence the results of clinical trials) (see Main point 1, and point 7).

This point appears to generally be a reiteration of the reviewer’s initial concerns on genotypic variation in the van den Heuvel et al., data set. We have addressed these concerns in response to points 1, 2 and 9.

Changes to Manuscript:

Please see changes in response to points 1,2 and 9 above.

12. Discussion – 6th paragraph: – 'Our model predicts <2.9% of single FSHD myocytes will be DUX4 target gene positive at any given time…' Should this be: '…DUX4 target gene-mRNA positive'. Also, does this 2.9%-figure differ between individual patients, and therefore might generally differ between FSHD1 and 2, and between different D4Z4 genetic categories ?; as this will matter for different individual patients.

We will update the manuscript to amend to ‘DUX4 target gene mRNA positive’. The reviewer again raises the concern of genotypic variability, we will of course discuss this and perform additional analysis as detailed above (points 1, 2, 9 and 11).

Changes to Manuscript:

Page 14, Line 526: We have made the following change to the Discussion:

‘Our model predicts <2.9% of single FSHD myocytes will be DUX4 target gene mRNA positive at any given time, despite significant DUX4 driven cell death.’

Please see additional changes in response to points 1,2 and 9 above, that also pertain to 11 and 12.

Reviewer #2 (Significance (Required)):Please see comments in the 'Evidence, reproducibility and clarity' box above.This paper is valuable in providing a mathematical basis for modelling the stochastic process of presence or absence of DUX4 expression in individual myocytes in FSHD (facioscapulohumeral muscular dystrophy), and hence a basis by which to study the effects of other factors or potential therapeutic interventions which may influence the overall proportion of muscle cells which do express DUX4, and hence succumb to its toxicity.This is a specialist field and the nature of presentation of the mathematical modelling in the paper will restrict accessibility to only a very few researchers in this field.

We appreciate that introducing concepts from different fields such as mathematics can present a challenge to some readers. However, new methods and interdisciplinary science are essential to understand complex disorders and this is even more so in complex rare diseases such as FSHD, where diversity in expertise is important. ‘Stochastic’ has been used to describe DUX4’s expression pattern for many years and is commonly used in many FSHD papers and conferences. We have introduced concepts from the well established field of ‘stochastic gene expression’ to understand DUX4 expression in FSHD. The methods are different from conventional approaches used in FSHD, but are arguably the most natural framework in which to understand DUX4.

We have provided user-friendly programs that can be used to estimate the effects on DUX4 parameters of multiple variables to make the paper useful, even if some of the more advanced mathematics is challenging. We will also further edit to increase accessibility.

Changes to Manuscript:

Multiple edits have been made in response to the helpful comments of both reviewers which has improved accessibility of the more technical aspects of the manuscript.

There appears to be a possible major problem in the way that the paper has combined data from patients with different genetic forms of FSHD, and different likely genetic severities. It would seem better to analyse the data separately for each of these to allow a more homogenous platform on which to construct a model.

That we pooled data from 4 FSHD patients, to derive estimates of 2 of the 5 parameters of our model is the major (non-editorial) comment about our study, and is raised in various versions in points 1, 2, 9, 11 and 12 of reviewer 2’s response. We have explained above that the reason for pooling is partly statistical, due to dataset size limitation. Considering each patient separately may prevent convergence of a gradient descent algorithm for parameter estimation. We again emphasise that pooling 4 FSHD patients, while less personalised, does not make our estimate of the average DUX4 transcription rate in FSHD patients invalid. It is simply an average level across our 4 patients, perhaps higher than for some patients but lower than for others.

As commented, we will repeat the analysis separately for the 2 patients who have larger numbers of DUX4 positive cells and report the findings, with the caveat that these estimates will, by definition, be over-estimates of the true value of the average DUX4 transcription rate for the FSHD population.

Changes to Manuscript:

Please see changes made in response to changes 1, 2 and 9 that also pertain to 11 and 12.